# Neurocircuitry-Inspired Hierarchical Graph Causal Attention Networks for Explainable Depression Identification

## Abstract

Major Depressive Disorder (MDD), affecting millions worldwide, exhibits complex pathophysiology manifested through disrupted brain network dynamics. Although graph neural networks that leverage neuroimaging data have shown promise in depression diagnosis, existing approaches are predominantly data-driven and operate largely as black-box models, lacking neurobiological interpretability. Here, we present NH-GCAT (Neurocircuitry-Inspired Hierarchical Graph Causal Attention Networks), a novel framework that bridges neuroscience domain knowledge with deep learning by explicitly and hierarchically modeling depression-specific mechanisms at different spatial scales. Our approach introduces three key technical contributions: (1) at the local brain regional level, we design a residual gated fusion module that integrates temporal blood oxygenation level dependent (BOLD) dynamics with functional connectivity patterns, specifically engineered to capture local depression-relevant low-frequency neural oscillations; (2) at the multi-regional circuit level, we propose a hierarchical circuit encoding scheme that aggregates regional node representations following established depression neurocircuitry organization, and (3) at the multi-circuit network level, we develop a variational latent causal attention mechanism that leverages a continuous probabilistic latent space to infer directed information flow among critical circuits, characterizing disease-altered whole-brain inter-circuit interactions. Rigorous leave-one-site-out cross-validation on the REST-meta-MDD dataset demonstrates NH-GCAT's state-of-the-art performance in depression classification, achieving a sample-size weighted-average accuracy of 73.3% and an AUROC of 76.4%, while simultaneously providing neurobiologically meaningful explanations. This work represents a significant advancement toward mechanism-aware, explainable artificial intelligence (AI) systems for psychiatric diagnosis.

## 1 Introduction

Major Depressive Disorder (MDD) is a leading cause of disability worldwide, with substantial individual and societal burden (Yan et al., 2019; Ferrari et al., 2013; Nestler et al., 2002). Early and accurate identification is critical for improving clinical outcomes, yet the neurobiological mechanisms underlying MDD remain poorly understood, posing significant challenges for objective diagnosis (Duman & Aghajanian, 2012; Drysdale et al., 2017). Recent advances in functional magnetic resonance imaging (fMRI) have enabled large-scale mapping of brain network dynamics, facilitating the identification of depression by analyzing altered functional connectivity patterns (Mulders et al., 2015). Since the brain network topology revealed by fMRI signals can be naturally described by graph models, Graph neural networks (GNNs) have shown promise for neuropsychiatric disorder classification (Ktena et al., 2017; Parisot et al., 2018; Bessadok et al., 2022; Wu et al., 2020; Isufi et al., 2021; Zheng et al., 2024c). By representing brain regions as nodes and their functional or structural relationships (like functional connectivity (FC)) as edges, GNNs can flexibly capture the topological and dynamic properties of brain networks. Typical GNN architectures stack multiple graph convolutional layers with generic message passing and aggregation schemes, followed by readout layers for classification, enabling the extraction of connectivity patterns relevant to MDD.

However, existing GNN methods suffer from limited accuracy and interpretability for MDD classification (Liu et al., 2024b; Zhao & Zhang, 2024). Their limited accuracy stems from treating brain regions uniformly and relying on static connectivity measures without considering critical temporal dynamics and MDD-specific alterations, leading to suboptimal feature extraction (Long et al., 2023; Ding, 2025; Calhoun et al., 2014). Their poor interpretability is primarily due to the lack of mechanisms for explicitly modeling the hierarchical organization of brain networks or the causal relationships between neural circuits that are fundamental to understanding depression (Zhao & Zhang, 2024; Long et al., 2023). Consequently, standard interpretability tools applied to these generic GNNs produce explanations that fail to align with established neuroscientific knowledge (Yu et al., 2024).

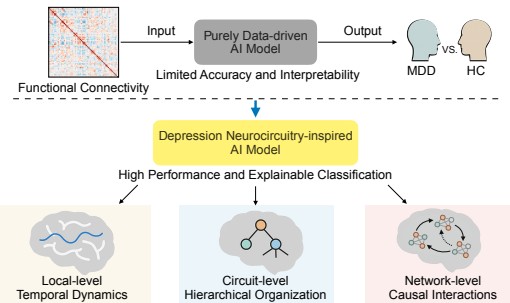

Figure 1: Comparison between conventional data-driven and our proposed neurocircuitry-inspired approaches for MDD classification. AI: Artificial Intelligence; MDD: Major Depressive Disorder; HC: Healthy Control.

Specifically, current neuroscience findings have indicated that depression pathophysiology manifests across multiple spatial scales of brain organization, each with distinct characteristics that present unique modeling challenges, as shown in Figure 1. At the local level, MDD patients exhibit altered temporal dynamics and frequency-specific neural oscillatory patterns, particularly in low-frequency bands associated with rumination and deficits in emotional processing (Ding, 2025; Calhoun et al., 2014). At the circuit level, dysfunctional integration within neural networks such as the default mode (DMN), salience (SN), frontoparietal (FPN), limbic (LN), and reward (RN) networks contributes to cognitive and emotional symptoms, with each circuit showing specific patterns of dysregulation (Johnson et al., 2024; Menon, 2011; Kaiser et al., 2015; Hamilton et al., 2011; Whitfield-Gabrieli & Ford, 2012; Noman et al., 2024). At the network level, aberrant causal relationships among the above circuits characterize the global dysregulation observed in MDD, with altered information flow and hierarchical control processes (Presigny & De Vico Fallani, 2022; Friston, 2011; Vidaurre et al., 2017; Morishima et al., 2025; Yeo et al., 2011; Csukly et al., 2024; Pearl, 2009). Developing computational models that incorporate structured neurobiological knowledge is crucial for improving both predictive performance and mechanistic interpretability in MDD identification (Von Rueden et al., 2021; Jiang et al., 2022; Munroe et al., 2024; Ali et al., 2023).

Here, we propose the Neurocircuitry-Inspired Hierarchical Graph Causal Attention Networks (NH-GCAT), a novel framework that bridges the gap between neuroscience and deep learning for explainable MDD identification. NH-GCAT systematically models depression-specific mechanisms across three spatial scales: 1) at the local brain regional level, we design a residual gated fusion (RG-Fusion) module that integrates temporal BOLD features with functional connectivity patterns, specifically engineered to capture depression-relevant low-frequency neural oscillations that conventional static FC approaches often overlook; 2) at the multi-regional circuit level, we propose a hierarchical circuit encoding scheme (HC-Pooling) that aggregates node representations following the established structure of depression-related circuits (DMN, FPN, SN, LN, RN). This biologically-informed operation enables modeling of dysregulated inter-regional communication, extraction of circuit-specific functional alterations, and interpretation of how local abnormalities propagate to network-level dysfunction, yielding features aligned with depression neurobiology; 3) at the multi-circuit network level, we develop a variational latent causal attention mechanism (VLCA) that leverages a continuous probabilistic latent space to infer directed information flow among critical circuits, characterizing disease-altered whole-brain inter-circuit interactions and providing mechanistic explanations for network-level dysfunctions in MDD.

Our contributions are summarized below: 1) We present a principled approach for integrating depression-specific neurocircuitry knowledge into GNN-based models; 2) We design novel modules (RG-Fusion, HC-Pooling and VLCA) for temporal dynamics integration, hierarchical aggregation and variational latent causal attention that enhance both predictive accuracy and interpretability; 3) We provide extensive empirical evidence that NH-GCAT not only achieves superior classification results but also uncovers mechanistic insights into MDD-related brain network alterations.

## 2 RELATED WORK

**Brain Network Identification and Interpretability for MDD.** Recent advances in graph neural networks (GNNs) have demonstrated promising results in brain network analysis (Ktena et al., 2017; Parisot et al., 2018; Yu et al., 2024; Dai et al., 2024). These approaches succeed in employing message passing mechanisms to capture region-wise interactions (Kang et al., 2024). For MDD classification specifically, existing GNNs primarily rely on static functional connectivity matrices as input features and treat brain regions as homogeneous nodes without considering their distinct neurobiological roles (Liu et al., 2024b; Zheng et al., 2024a). While some recent works (Kong et al., 2025; Zhao & Zhang, 2024) attempt to incorporate temporal information through sequence modeling, they often fail to effectively capture the low-frequency oscillatory patterns that are crucial for depression diagnosis.

Current interpretable approaches in neuroimaging broadly fall into two categories: post-hoc explanation methods and architecture-constrained models. Post-hoc methods, including attention visualization and feature attribution (Zheng et al., 2024c; Rudin, 2019; Zhang et al., 2023; Sundararajan et al., 2017; Veličković et al., 2018), provide limited insight into neurobiological mechanisms. Architecture-constrained approaches incorporate anatomical priors (Von Rueden et al., 2021; Zheng et al., 2024b; Liu et al., 2024a; Jiang et al., 2020), but typically treat these as static constraints rather than modeling dynamic disease processes.

**Techniques for Neural Circuit Modeling.** Residual gating mechanisms have demonstrated success in natural language processing (Tai et al., 2015; Greff et al., 2016; Choi et al., 2018) and time series analysis (Bresson & Laurent, 2017; Chen et al., 2019; Afzal et al., 2024), allowing models to selectively integrate information streams. When applied to neural time series data such as EEG, approaches like HybGNN (Wang et al., 2024) effectively capture temporal dynamics. However, unlike EEG's high temporal resolution, fMRI analysis requires modeling specific low-frequency BOLD oscillations for MDD, necessitating specialized fusion mechanisms beyond standard sequence modeling. Dynamic functional connectivity (Damaraju et al., 2014) and frequency-specific neural oscillations (Tadayonnejad et al., 2016) have been extensively investigated in fMRI research; however, their integration with graph neural networks remains limited.

Hierarchical representation learning in graph structures has shown significant utility across domains including molecular property prediction and social network analysis (Ying et al., 2018). Recent Transformer-based methods have begun to leverage community structures in brain networks. For instance, Com-BrainTF (Bannadabhavi et al., 2023) and BrainGT (Shehzad et al., 2024) utilize prompt tokens or dual-attention to capture functional communities, while BioBGT (Peng et al., 2025) and THC (Dai et al., 2023) employ spectral entropy or data-driven clustering to encode small-world properties. Most approaches, however, employ generic clustering objectives rather than leveraging domain-specific organizational principles. In contrast to these data-driven or soft-attention methods, our approach explicitly enforces a bottom-up aggregation based on established depression neurocircuitry to ensure mechanistic interpretability. In neuroscience, hierarchical approaches have been applied to structural brain networks and functional parcellations (Csukly et al., 2024; Liu et al., 2024a; Jiang et al., 2020), but rarely incorporate established circuit-level knowledge. The potential to align hierarchical graph representations with known neurocircuitry organization could significantly improve both model performance and interpretability in MDD identification.

Variational approaches for inferring latent graph structures (Sanchez-Martin et al., 2021; Bahuleyan et al., 2017) and disentangled representations (Jeong & Song, 2019; Yang et al., 2021) have shown success in uncovering hidden relationships in complex data. Causal methods such as dynamic causal modeling (Friston, 2011; Pearl, 2009) and Granger causality (Seth et al., 2015) provide frameworks for understanding information flow. Notably, BrainOOD (Xu et al., 2025) proposes causal subgraph learning for out-of-distribution generalization (invariant learning). While BrainOOD aims to remove environmental bias, our work focuses on inferring effective connectivity (directed information flow) to explain pathophysiological mechanisms. Existing approaches that model causality in graph-structured data (Sanchez-Martin et al., 2021; Behnam & Wang, 2024; Wang et al., 2023; Sui et al., 2022; Wang et al., 2022) primarily focus on region-level interactions, leaving circuit-level causal relationships - which align better with neuroscientific theories of depression - relatively unexplored. The integration of probabilistic causal modeling with circuit-level analysis represents a promising avenue for advancing mechanistic understanding of MDD.

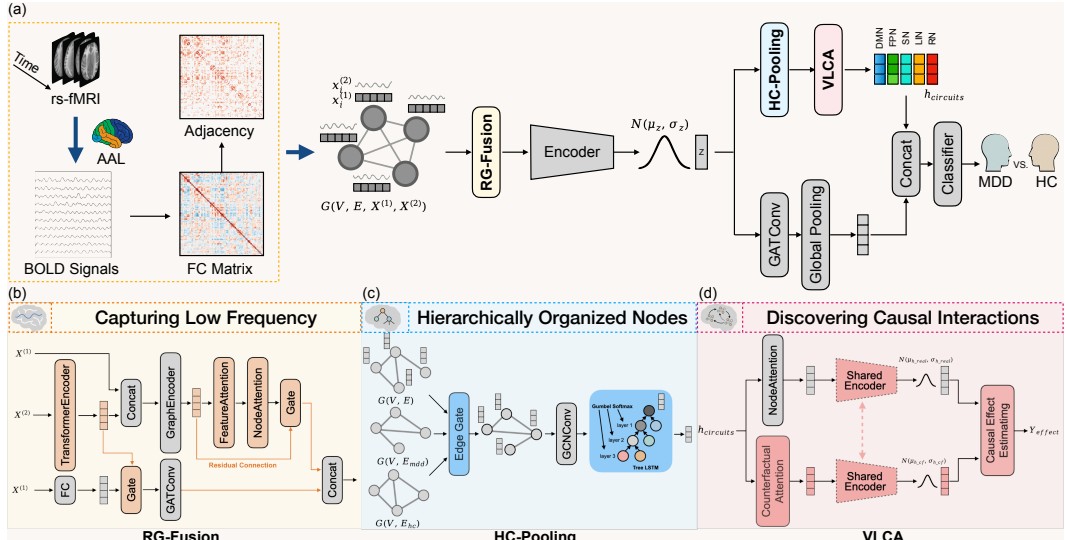

Figure 2: The overall framework of the proposed NH-GCAT. BOLD: Blood Oxygenation Level Dependent; FC: Functional Connectivity; MDD: Major Depressive Disorder; HC: Healthy Control.

## 3 METHODOLOGY

We present NH-GCAT (Neurocircuitry-Inspired Hierarchical Graph Causal Attention Networks), a novel framework that integrates neuroscientific domain knowledge with deep learning for explainable and accurate depression classification. As illustrated in Figure 2, NH-GCAT comprises three principal components: 1) RG-Fusion: a residual gated fusion module for integrating temporal BOLD dynamics with functional connectivity patterns, 2) RC-Pooling: a hierarchical circuit encoding scheme that aggregates node representations according to established depression neurocircuitry, and 3) VLCA: a variational latent causal attention mechanism for modeling and interpreting inter-circuit interactions. The rationale behind these key architectural design choices is further elaborated on Appendix A.3.

**Problem Formulation.** Given resting-state fMRI (rs-fMRI) data from $N$ subjects, our objective is to classify each subject as either major depressive disorder (MDD) or healthy control (HC). For each subject $i$, we obtain a static feature matrix $\mathbf{X}_i^{(1)} \in \mathbb{R}^{n \times m}$, which includes the functional connectivity (FC) matrix computed as the pairwise Pearson correlation between BOLD signals, as well as clinical variables such as age, sex, and education. Additionally, we have a time series matrix of BOLD signals $\mathbf{X}_i^{(2)} \in \mathbb{R}^{n \times T}$, where $n$ is the number of brain regions (ROIs) and $T$ is the number of time points. Each subject is assigned a binary label $y^{(i)} \in 0, 1$, indicating HC (0) or MDD (1). We represent each subject's brain as a graph $\mathcal{G}_i = (\mathcal{V}_i, \mathcal{E}_i, \mathbf{X}_i^{(1)}, \mathbf{X}_i^{(2)})$, where $\mathcal{V}_i$ denotes the set of ROIs and $\mathcal{E}_i$ encodes the functional connection based edges. The goal is to learn a function $f : \mathcal{G} \to 0, 1$ that achieves accurate classification and provides interpretable, neuroscientifically meaningful explanations.

**Residual Gated Fusion for Temporal Dynamics Integration (RG-Fusion).** The RG-Fusion module is designed to effectively integrate complementary information from both static functional connectivity patterns and temporal BOLD dynamics. This integration is crucial for capturing the full spectrum of neural activity characteristics in rs-fMRI data, particularly the low-frequency fluctuations that are clinically significant in depression neuroimaging. The RG-Fusion module processes $\mathbf{X}^{(1)}$ and $\mathbf{X}^{(2)}$ through parallel pathways as follows:

**Temporal Feature Processing.** The temporal BOLD signals $\mathbf{X}^{(2)}$ are processed through a transformer encoder to capture global dependencies:

$$\mathbf{H}_{\text{temp}} = \text{TransformerEncoder}(\mathbf{X}^{(2)}) \in \mathbb{R}^{n \times d} \tag{1}$$

where $d$ is the latent dimension. The output is further concatenated with the original static features $\mathbf{X}^{(1)}$ and then refined using GraphEncoder module, which dual-path graph convolutions (SAGE-

Conv and GATConv), to capture both local and global topological properties:

$$\mathbf{Z}_{\text{temp}} = \text{GraphEncoder}(\mathbf{H}_{\text{temp}}, \mathcal{E}) \in \mathbb{R}^{n \times d} \tag{2}$$

The processing of GraphEncoder can be formulated as:

$$\mathbf{H}_2 = \text{Concat}(\mathbf{X}^{(1)}, \mathbf{H}_{\text{temp}}) \in \mathbb{R}^{n \times d'} \tag{3}$$

$$\mathbf{Z}_{\text{temp}} = \text{Concat}(\text{SAGEConv}(\mathbf{H}_2, \mathcal{E}), \ \text{GATConv}(\mathbf{H}_2, \mathcal{E})) \in \mathbb{R}^{n \times d'} \tag{4}$$

**Static Feature Processing.** Simultaneously, the static features $\mathbf{X}^{(1)}$ are processed through fully connected layers (MLP) followed by Gate module and graph attention convolution:

$$\mathbf{Z}_{\text{static}} = \text{GATConv}(\text{Gate}(\text{MLP}(\mathbf{X}^{(1)}), \mathbf{H}_{\text{temp}}), \mathcal{E}) \in \mathbb{R}^{n \times d} \tag{5}$$

**Gate Module.** The Gate module leverages an adaptive gating mechanism to selectively integrate information from both pathways, ensuring that the distinctive characteristics of each are effectively retained and utilized:

$$\mathbf{G} = \sigma\left(\mathbf{W}_g\left[\mathbf{Z}_1 \mid \mathbf{Z}_2\right] + \mathbf{b}_g\right) \in \mathbb{R}^{n \times d}, \quad \mathbf{Z}_{\text{fused}} = \mathbf{G} \odot \mathbf{Z}_1 + (1 - \mathbf{G}) \odot \mathbf{Z}_2 \in \mathbb{R}^{n \times d} \tag{6}$$

where $\sigma$ is the sigmoid activation function, $\mathbf{W}_g$ and $\mathbf{b}_g$ are learnable parameters, $\mid$ denotes concatenation, and $\odot$ represents element-wise multiplication. $\mathbf{Z}_1$ and $\mathbf{Z}_2$ denote the two feature vectors that need to be fused.

**Residual Connection.** We enhance feature discriminability through a hierarchical two-stage attention mechanism coupled with residual gated fusion. First, FeatureAtttention adaptively weights temporal features for each node, followed by NodeAtttention which focuses on depression-relevant brain regions. The attended features $\mathbf{H}_{\text{attn}}$ are then combined with $\mathbf{Z}_{\text{temp}}$ via residual gating, and integrated with $\mathbf{Z}_{\text{static}}$ to produce the final representation $\mathbf{Z}_{\text{final}}$.

$$\mathbf{H}_{\text{final}} = \text{Gate}\left(\mathbf{Z}_{\text{temp}}, \mathbf{H}_{\text{attn}}\right) \in \mathbb{R}^{n \times d}, \quad \mathbf{Z}_{\text{final}} = \text{Concat}\left(\mathbf{H}_{\text{final}}, \mathbf{Z}_{\text{static}}\right) \in \mathbb{R}^{n \times d'} \tag{7}$$

$\mathbf{Z}_{\text{final}}$ is transformed through a variational encoder to obtain $\mathbf{Z}_{\text{ve}}$, yielding a continuous latent representation that encapsulates both static network properties and dynamic temporal characteristics of brain activity, providing a comprehensive embedding for subsequent modules in the NH-GCAT framework.

**Hierarchical Circuit Encoding (HC-Pooling).** To incorporate neurobiological priors, we design a hierarchical circuit encoding scheme that aggregates node representations according to the established organization of depression-related neural circuits.

**Circuit-specific Node Assignment.** Let $\mathcal{C} = c_1, ..., c_5$ be depression-related circuits (DMN, SN, FPN, LN, RN). For each circuit $c_j$, we define $\mathcal{V}_{c_j} \subset \mathcal{V}$ as its constituent regions based on neuroanatomical knowledge.

**Adjacency Reconstruction.** For $\mathcal{V}_{c_j}$, we derive FC matrix $\mathbf{A}^{(c_j)}$ by fusing subject and group-level FC priors via a learnable gating mechanism:

$$\mathbf{A}^{(c_j)} = \sum_{k=1}^{3} \text{softmax}(\text{MLP}(\mathbf{Z}_{\text{ve}}^{(c_j)})) \cdot \mathbf{A}_k \tag{8}$$

where $\mathbf{A}_1$, $\mathbf{A}_2$, and $\mathbf{A}_3$ represent individual functional connectivity, MDD group-level average connectivity, and HC group-level average connectivity matrices, respectively.

**Top-down Hierarchical Organization.** For each neural circuit $c_j$, we employ a differentiable top-down hierarchical organization approach using Gumbel-Softmax to assign nodes to different hierarchical levels. First, we compute node embeddings using a Graph Convolutional Network:

$$\mathbf{H}^{(c_j)} = \text{GCN}(\mathbf{Z}_{\text{ve}}^{(c_j)}, \mathbf{A}^{(c_j)}) \tag{9}$$

We then assign nodes to three hierarchical levels using differentiable masks:

$$\begin{aligned} \mathbf{M}_1 &= \text{GumbelSoftmax}\big(f_1(\mathbf{H}), \tau\big), \\ \mathbf{M}_2 &= \text{GumbelSoftmax}\big(f_2(\mathbf{H}), \tau, \text{mask} = (\mathbf{M}_1 < \epsilon)\big), \quad \mathbf{M}_3 = 1 - \mathbf{M}_1 - \mathbf{M}_2 \end{aligned} \tag{10}$$

where $f_1$ and $f_2$ are linear projection, $\tau$ is the temperature parameter for Gumbel-Softmax, and $\epsilon$ is a small threshold.

**Bottom-up Hierarchical Aggregation.** We employ a ChildSumTreeLSTM (Tai et al., 2015) to aggregate information from lower to higher hierarchical levels. For each level $l \in 3, 2, 1$, we compute:

$$\mathbf{h}_l = \mathbf{H} \odot \mathbf{M}_l, \quad \mathbf{c}_l = \mathbf{C} \odot \mathbf{M}_l \tag{11}$$

where $\mathbf{H}$ and $\mathbf{C}$ are the hidden and cell states, $\odot$ represents element-wise multiplication. Bottom-up aggregation proceeds as follows:

$$
\begin{aligned}
\mathbf{h}_{\text{low}}, \mathbf{c}_{\text{low}} &= \text{ChildSumTreeLSTM}(\mathbf{h}_3, \mathbf{c}_3, \mathbf{M}_3) \\
\mathbf{h}_{\text{mid}}, \mathbf{c}_{\text{mid}} &= \text{ChildSumTreeLSTM}([\mathbf{h}_{\text{low}}, \mathbf{h}_2], [\mathbf{c}_{\text{low}}, \mathbf{c}_2], \mathbf{M}_2) \\
\mathbf{h}_{\text{root}}, \mathbf{c}_{\text{root}} &= \text{ChildSumTreeLSTM}([\mathbf{h}_{\text{mid}}, \mathbf{h}_1], [\mathbf{c}_{\text{mid}}, \mathbf{c}_1], \mathbf{M}_1)
\end{aligned}
\tag{12}
$$

where $\mathbf{h}$ and $\mathbf{c}$ represent the hidden and cell states from the TreeLSTM, respectively. The subscripts 'low' and 'mid' denote the intermediate aggregated states from the lower and middle hierarchical levels. The final state, $\mathbf{h}_{\text{root}}$, denotes the comprehensive circuit-level embedding. Accordingly, the HC-Pooling module produces $\mathbf{H}_{\text{DMN}}, \mathbf{H}_{\text{SN}}, \mathbf{H}_{\text{FPN}}, \mathbf{H}_{\text{LN}}, \mathbf{H}_{\text{RN}}$, corresponding to the aggregated embeddings of the DMN, SN, FPN, LN, and RN circuits, respectively. The ChildSumTreeLSTM operation is defined as:

$$
\begin{aligned}
\mathbf{i} &= \sigma\left(\mathbf{W}_i \, \mathbf{h}_{\text{sum}} + \mathbf{U}_i \, \mathbf{h}_{\text{sum}}\right), & \mathbf{o} &= \sigma\left(\mathbf{W}_o \, \mathbf{h}_{\text{sum}} + \mathbf{U}_o \, \mathbf{h}_{\text{sum}}\right), \\
\mathbf{u} &= \tanh\left(\mathbf{W}_u \, \mathbf{h}_{\text{sum}} + \mathbf{U}_u \, \mathbf{h}_{\text{sum}}\right), & \mathbf{f}_k &= \sigma\left(\mathbf{U}_f \, \mathbf{h}_k\right), \\
\mathbf{c} &= \mathbf{i} \odot \mathbf{u} + \sum_{k \in \mathcal{N}} \mathbf{f}_k \odot \mathbf{c}_k, & \mathbf{h} &= \mathbf{o} \odot \tanh(\mathbf{c})
\end{aligned}
\tag{13}
$$

where $\mathbf{h}_{\text{sum}} = \sum k \in \mathcal{N} \mathbf{h}_k$ is the sum of child node representations, $\mathcal{N}$ is the set of child nodes, and $\sigma$ is the sigmoid function. To guide the model toward learning clinically relevant connectivity patterns, we constrain the learned adjacency matrix using group-level priors:

$$\mathcal{L}_{\text{mse}} = |\mathbf{A}^{(c_j)} - \mathbf{A}_{y_i}|_F^2 \tag{14}$$

where $\mathbf{A}_{y_i}$ represents the group-level connectivity prior corresponding to subject $i$'s label.

**Variational Latent Causal Attention (VLCA).** To model causal interactions between neural circuits and provide mechanistic explanations for depression, we introduce VLCA, which enables counterfactual reasoning about circuit-level interactions. Given circuit-level embeddings $\mathbf{H}_{\text{DMN}}, \mathbf{H}_{\text{SN}}, \mathbf{H}_{\text{FPN}}, \mathbf{H}_{\text{LN}}, \mathbf{H}_{\text{RN}} \in \mathbb{R}^{B \times d}$, VLCA first computes attention-weighted interactions:

$$\mathbf{Q}, \mathbf{K}, \mathbf{V} = \mathbf{W}_q \mathbf{H}, \mathbf{W}_k \mathbf{H}, \mathbf{W}_v \mathbf{H}, \quad \mathbf{A}^{\text{real}} = \text{softmax}\left(\frac{\mathbf{Q}\mathbf{K}^T}{\sqrt{d}}\right), \quad \mathbf{H}^{\text{real}} = \mathbf{A}^{\text{real}}\mathbf{V} \tag{15}$$

where $\mathbf{H} \in \mathbb{R}^{B \times C \times d}$ represents stacked circuit embeddings, and $\mathbf{A}^{\text{real}}$ captures circuit interactions. Then the attention-weighted representations are encoded into a continuous latent space:

$$\boldsymbol{\mu}^{\text{real}}, \log \boldsymbol{\sigma}^{2\text{real}} = f_{\text{encoder}}(\mathbf{H}^{\text{real}}), \quad \mathbf{z}^{\text{real}} = \boldsymbol{\mu}^{\text{real}} + \boldsymbol{\sigma}^{\text{real}} \odot \boldsymbol{\epsilon}, \quad \boldsymbol{\epsilon} \sim \mathcal{N}(0, \mathbf{I}) \tag{16}$$

where $f_{\text{encoder}}$ is a neural network. For counterfactual reasoning, we replace learned attention with an identity matrix (self-attention only):

$$\mathbf{A}^{\text{cf}} = \mathbf{I}_C, \quad \mathbf{H}^{\text{cf}} = \mathbf{A}^{\text{cf}}\mathbf{V} \tag{17}$$

Using the same encoder with shared parameters:

$$\boldsymbol{\mu}^{\text{cf}}, \log \boldsymbol{\sigma}^{2\text{cf}} = f_{\text{encoder}}(\mathbf{H}^{\text{cf}}), \quad \mathbf{z}^{\text{cf}} = \boldsymbol{\mu}^{\text{cf}} + \boldsymbol{\sigma}^{\text{cf}} \odot \boldsymbol{\epsilon}' \tag{18}$$

The causal effect of circuit interactions is estimated as:

$$\mathbf{y}^{\text{effect}} = f_{\text{pred}}(\mathbf{z}^{\text{real}}) - f_{\text{pred}}(\mathbf{z}^{\text{cf}}) \tag{19}$$

Our learning objective combines classification loss on the causal effect with KL regularization:

$$\mathcal{L}_{\text{VLCA}} = \mathcal{L}_{\text{CE}}(\mathbf{y}^{\text{effect}}, \mathbf{y}) + \beta \mathcal{D}_{\text{KL}}(\mathcal{N}(\boldsymbol{\mu}^{\text{real}}, \boldsymbol{\sigma}^{2\text{real}}) \| \mathcal{N}(\boldsymbol{\mu}_{\text{prior}}, \mathbf{I})) \tag{20}$$

where $\mathcal{L}_{\text{CE}}$ is the cross-entropy loss, $\mathcal{D}_{\text{KL}}$ is the Kullback-Leibler divergence, and $\boldsymbol{\mu}_{\text{prior}}$ is either zero or the mean of the input features depending on the prior type. This formulation enables the model to learn interpretable circuit interaction patterns, quantify their causal effect on depression

classification, and provide insights into how altered circuit communication contributes to MDD pathophysiology.

**Training Objective.** Our overall training objective combines multiple loss terms to balance classification performance, representation learning, and causal understanding:

$$\mathcal{L} = \mathcal{L}_{\text{cls}} + \lambda_{\text{kl}}\mathcal{L}_{\text{kl}} + \lambda_{\text{VLCA}}\mathcal{L}_{\text{VLCA}} + \lambda_{\text{mse}}\mathcal{L}_{\text{mse}} \tag{21}$$

where $\mathcal{L}_{\text{cls}}$ denotes the cross-entropy loss for MDD classification, $\mathcal{L}_{\text{kl}}$ represents the KL divergence regularization from the backbone's variational encoding, and hyperparameters $\lambda_{\text{kl}}$, $\lambda_{\text{VLCA}}$, and $\lambda_{\text{mse}}$ balance these competing objectives.

## 4 EXPERIMENTS

### 4.1 EXPERIMENTAL SETTINGS

**Datasets and Preprocessing.** We utilized the REST-meta-MDD dataset, comprising 1,601 participants (830 MDD, 771 HC) from 16 sites after rigorous quality control procedures (Yan et al., 2019; Chen et al., 2022). We extracted BOLD time series from 116 regions using the AAL atlas (Tzourio-Mazoyer et al., 2002), computed Fisher z-transformed functional connectivity, and constructed brain graphs using k-nearest neighbors (k=40). Population-level reference graphs were generated for MDD and HC groups to provide connectivity templates for the hierarchical circuit encoding. Details in Appendix A.1.

**Baselines and Evaluation.** We compared NH-GCAT against general-purpose graph architectures (GAT (Veličković et al., 2018), GIN (Xu et al., 2018), GraphSAGE (Hamilton et al., 2017), GPS (Rampášek et al., 2022), GCN (Kipf & Welling, 2016)) and state-of-the-art MDD classification methods (BrainIB (Zheng et al., 2024c), CI-GNN (Zheng et al., 2024a), LCCAF (Kang et al., 2024), etc.). Performance was evaluated using accuracy (ACC), area under the ROC curve (AUC), F1-score, sensitivity (SEN), and specificity (SPE), with 5-fold and leave-one-site-out cross-validation protocols. Details in Appendix A.2.

**Implementation.** Our model used 128-dimensional hidden layers with a 64-dimensional single-head causal attention mechanism. We employed Adam optimizer with gradient clipping and dynamic weight scheduling for regularization terms. All experiments are implemented using the PyTorch framework, and computations are performed on one NVIDIA RTX 4090 GPU. More details can be found in Appendix A.4.

### 4.2 PERFORMANCE COMPARISON

**Overall Classification Results.** Table 1 presents a comprehensive comparison between our proposed NH-GCAT model and a range of state-of-the-art methods and strong baselines on the REST-meta-MDD dataset. NH-GCAT achieves the highest performance across four out of five metrics, demonstrating its effectiveness for MDD classification. Specifically, NH-GCAT attains an AUC of 78.5% (1.7), accuracy of 73.8% (1.4), specificity of 71.0% (6.6), and F1-score of 75.0% (1.8), substantially outperforming competing models in these key metrics. Notably, NH-GCAT surpasses the previous best AUC (75.6%) from LCCAF (Kang et al., 2024) by a significant margin of +2.9%, and improves upon the strongest accuracy (73.0%) of BPI-GNN (Zheng et al., 2024b) by +0.8%. The F1-score exhibits a substantial gain of +2.4% over the best competing method (LGMF-GNN). For specificity, NH-GCAT achieves 71.0%, representing a modest improvement of +0.3% over the previous best (LCCAF, 70.7%). While NH-GCAT achieves the second-best sensitivity at 76.4%, it falls short of GAT-Baseline's 77.5% by only 1.1%, indicating competitive performance in detecting MDD cases. Furthermore, we observe that external models exhibit inconsistent performance across metrics. For instance, while LCCAF achieves competitive AUC and specificity, it shows substantial variation in accuracy (70.2% ± 8.3%). Among our implemented baselines, GAT-Baseline achieves the highest sensitivity but suffers from poor specificity (57.2%), indicating a significant trade-off between correctly identifying positive and negative cases. In contrast, NH-GCAT maintains balanced and robust performance across all metrics, with consistently low standard deviations, demonstrating its stability and reliability for clinical applications where both high sensitivity and specificity are crucial for accurate diagnosis.

Table 1: Comprehensive performance comparison with state-of-the-art methods and baselines for MDD classification on REST-meta-MDD dataset. The best results are marked in bold and the second-best value is underlined. The standard deviations are in parentheses. Improvement shows the performance gain of NH-GCAT over the best competing method for each metric.

| Model | AUC | ACC | SEN | SPE | F1 |
|---|---|---|---|---|---|
| *External models* | | | | | |
| BrainIB (Zheng et al., 2024c) | - | 70.0 (2.2) | - | - | - |
| MV-GNN (Zhang et al., 2023) | 66.6 (5.2) | 65.6 (4.3) | 63.4 (11.2) | - | 64.6 (6.0) |
| GC-GAN (Oh et al., 2024) | - | 66.8 (4.3) | 70.2 (7.9) | 63.1 (8.4) | 68.7 (4.6) |
| DSFGNN (Zhao & Zhang, 2024) | 71.6 | 67.1 | 65.4 | - | 67.3 |
| BPI-GNN (Zheng et al., 2024b) | - | 73.0 (1.0) | - | - | 72.0 (1.0) |
| TEM (Dai et al., 2024) | 70.7 | 68.6 | 69.8 | 67.9 | - |
| CI-GNN (Zheng et al., 2024a) | - | 72.0 (2.0) | - | - | 70.0 (1.0) |
| LGMF-GNN (Liu et al., 2024b) | 73.7 (2.7) | 71.3 (1.5) | 73.5 (6.3) | - | 72.6 (2.1) |
| BrainNPT (Hu et al., 2024) | 70.6 (3.5) | 66.7 (3.6) | - | - | - |
| MSSTAN (Kong et al., 2025) | 67.1 (1.4) | 68.7 (9.0) | 74.7 (3.3) | 59.5 (4.8) | 71.6 (1.2) |
| LCCAF (Kang et al., 2024) | 75.6 (1.0) | 70.2 (8.3) | 69.7 (2.7) | 70.7 (2.1) | - |
| *Our implemented baselines* | | | | | |
| GCN | 70.6 (2.4) | 65.8 (1.1) | 67.2 (10.0) | 64.2 (10.1) | 66.8 (4.0) |
| GIN | 70.8 (2.0) | 66.3 (1.9) | 65.7 (14.4) | 67.0 (12.7) | 66.3 (5.2) |
| GraphSAGE | 69.8 (2.6) | 65.7 (1.5) | 64.1 (7.4) | 67.3 (8.5) | 65.8 (2.8) |
| GPS | 67.6 (5.0) | 64.3 (3.9) | 63.3 (16.4) | 65.5 (10.9) | 63.9 (8.4) |
| GAT-Baseline | 71.5 (3.2) | 67.7 (2.7) | **77.5 (9.1)** | 57.2 (9.4) | 71.2 (3.3) |
| **NH-GCAT (Ours)** | **78.5 (1.7)** | **73.8 (1.4)** | 76.4 (5.8) | **71.0 (6.6)** | **75.0 (1.8)** |
| Improvement | +2.9 | +0.8 | -1.1 | +0.3 | +2.4 |

Table 2: Leave-one-site-out cross-validation accuracy (%) for MDD classification across 16 sites on REST-meta-MDD dataset. MDD and HC indicate sample sizes per site. The final column shows the sample-size weighted average (W. Avg.).

| | Site | | | | | | | | | | | | | | | | W. Avg. |
|---|---|---|---|---|---|---|---|---|---|---|---|---|---|---|---|---|---|
| | 1 | 2 | 3 | 4 | 5 | 6 | 7 | 8 | 9 | 10 | 11 | 12 | 13 | 14 | 15 | 16 | |
| MDD | 73 | 16 | 35 | 54 | 48 | 45 | 20 | 20 | 61 | 30 | 41 | 18 | 250 | 79 | 18 | 22 | |
| HC | 73 | 14 | 62 | 48 | 26 | 17 | 16 | 32 | 37 | 41 | 31 | 229 | 65 | 20 | 23 | | |
| CI-GNN (Zheng et al., 2024a) | 63.0 | **83.0** | 76.0 | 70.0 | 81.0 | **75.0** | 73.0 | 72.0 | 68.0 | 81.0 | **75.0** | 73.0 | 63.0 | 68.0 | **75.0** | 64.0 | 69.2 |
| BrainIB (Zheng et al., 2024c) | 63.3 | 73.0 | **77.8** | 71.3 | 68.8 | 73.2 | 75.7 | **80.6** | 72.0 | **82.1** | 67.1 | 69.4 | 63.2 | **70.1** | 71.1 | 68.9 | 68.8 |
| NH-GCAT | **67.8** | 80.0 | 72.2 | **72.4** | **83.3** | 71.8 | **86.5** | 69.4 | **72.0** | 74.6 | 69.5 | **81.6** | **73.3** | 68.8 | 73.7 | **77.8** | **73.3** |
| Improvement | +4.5 | -3.0 | -5.6 | +1.1 | +2.3 | -3.2 | +10.8 | -11.2 | 0.0 | -7.5 | -5.5 | +8.6 | +10.1 | -1.3 | -1.3 | +8.9 | +4.1 |

**Leave-One-Site-Out Generalization.** Table 2 shows the leave-one-site-out cross-validation (LOSO-CV) accuracy for NH-GCAT, CI-GNN, and BrainIB across 16 sites. NH-GCAT consistently achieves higher or competitive accuracy on most sites, with a sample-size weighted-average accuracy of 73.3%, outperforming both CI-GNN (69.2%) and BrainIB (68.8%). Specifically, NH-GCAT attains the highest accuracy on 8 out of 16 sites, and achieves notable improvements (e.g., +10.8% and +10.1%) on sites 7 and 13, respectively. Nevertheless, it underperforms on a few sites (e.g., sites 2, 3, 6, 8, 10, 11, 14, 15), which may be attributed to site-specific variations such as data imbalance or heterogeneity in acquisition protocols. Despite these fluctuations, the overall improvement in weighted-average accuracy (+4.1% over CI-GNN and +4.5% over BrainIB) demonstrates the robustness and generalizability of NH-GCAT across diverse clinical sites. Site-specific performance are provided in Appendix A.6.

## 4.3 ABLATION STUDY

Table 3 quantifies each component's contribution to NH-GCAT's performance. The RG-Fusion module improves AUC (+3.3%) and accuracy (+2.5%) over the GAT baseline, with a notable increase in specificity (+13.4%), demonstrating the value of integrating temporal BOLD dynamics with static functional connectivity. Adding VLCA further enhances AUC (+1.1%), accuracy (+1.8%), and F1 score (+3.1%), confirming the importance of modeling causal circuit interactions. The complete model with HC-Pooling achieves statistically significant improvements over the baseline in AUC (+7.0%), accuracy (+6.1%), specificity (+13.8%), and F1 score (+3.8%), while maintaining competitive sensitivity performance. These results validate our neurocircuitry-inspired design choices and their contributions to MDD classification. More details in Appendix A.7.

Table 3: Ablation study showing the contribution of each component in the NH-GCAT framework. The best results are marked in bold and the standard deviations are in parentheses. *Increment* rows show the performance change after adding each component. *Statistically significant improvement over GAT-Baseline ($p < 0.05$, Wilcoxon signed-rank test).

| Model Variant | AUC | ACC | SEN | SPE | F1 |
|---|---|---|---|---|---|
| GAT-Baseline | 71.5 (3.2) | 67.7 (2.7) | **77.5 (9.1)** | 57.2 (9.4) | 71.2 (3.3) |
| + RG-Fusion | 74.8 (2.3) | 70.2 (1.7) | 69.9 (10.9) | 70.6 (9.9) | 70.5 (4.3) |
| *Increment* | *+3.3* | *+2.5* | *-7.6* | *+13.4* | *-0.7* |
| + VLCA | 75.9 (2.0) | 72.0 (2.0) | 75.4 (5.4) | 68.2 (6.5) | 73.6 (2.1) |
| *Increment* | *+1.1* | *+1.8* | *+5.5* | *-2.4* | *+3.1* |
| + HC-Pooling (Full) | **78.5 (1.7)** * | **73.8 (1.4)** * | 76.4 (5.8) | **71.0 (6.6)** * | **75.0 (1.8)** * |
| *Increment* | *+2.6* | *+1.8* | *+1.0* | *+2.8* | *+1.4* |
| **Total Improvement** | **+7.0** | **+6.1** | **-1.1** | **+13.8** | **+3.8** |

## 4.4 INTERPRETABILITY ANALYSIS

NH-GCAT provides neuroscientifically meaningful explanations for MDD pathophysiology through three complementary analyses (Figure 3).

**Frequency-specific Neural Dynamics.** We validated our RG-Fusion module by separately feeding low-frequency (0.01–0.08 Hz) and high-frequency (0.1–0.25 Hz) BOLD signals into the trained model. Our RG-Fusion module shows significantly higher AUC with low-frequency inputs (0.742±0.019) versus high-frequency inputs (0.679±0.032) ($p = 0.0037$). This confirms that our model captures depression-relevant neural oscillations predominantly manifested in low-frequency BOLD dynamics, as shown in Figure 3(a).

**Hierarchical Circuit Organization.** Figure 3(b) visualizes our HC-Pooling module's assignment of brain regions to three hierarchical layers across neural circuits (Layer-1: high-level integration, Layer-2: intermediate processing, Layer-3: primary processing). Statistical analysis revealed significant MDD-HC differences in key regions across circuits, including Angular_L, Frontal_Sup_Medial_L (FSM_L), Frontal_Inf_Oper_R (FIO_R), Amygdala_R, ParaHippocampal_R (PHC_R), and Caudate_L. MDD exhibits: (1) increased high-level representation in DMN regions (FSM_L, Angular_L), consistent with pathological rumination; (2) reduced high-level representation in frontoparietal regions (FIO_R), suggesting impaired cognitive control; (3) increased low-level representation in limbic regions (Amygdala_R), indicating less regulated emotional processing; and (4) altered hierarchical organization in reward network regions (Caudate_L), potentially reflecting compensatory mechanisms for reward deficits. More details can be found in Appendix A.8.2.

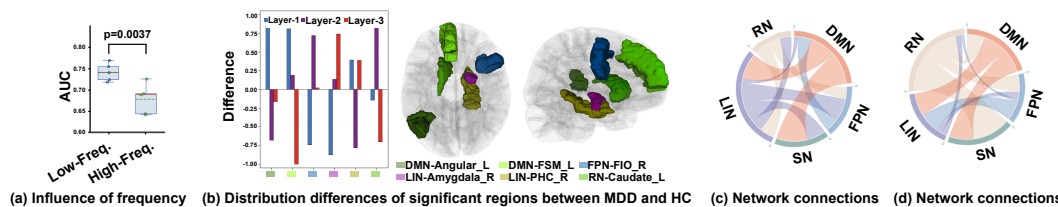

(a) Influence of frequency   (b) Distribution differences of significant regions between MDD and HC   (c) Network connections in HC   (d) Network connections in MDD

Figure 3: Multi-scale interpretability analysis of NH-GCAT for MDD classification.

**Causal Inter-circuit Interactions.** The VLCA mechanism reveals distinct patterns of information flow among neural circuits in MDD versus HC groups, visualized in Figure 3(c-d). Quantitative analysis of these network connections shows MDD exhibits: (1) DMN receives abnormally increased input from reward networks, suggesting pathological integration of reward signals into self-referential processing—potentially underlying negative reward prediction errors and rumination; (2) SN receives reduced regulatory input from DMN, indicating impaired top-down modulation of salience detection by self-referential processes; (3) LIN receives diminished regulatory signals from DMN, reflecting weakened control over emotional reactivity; (4) LIN receives novel regulatory input from FPN, suggesting emergence of compensatory top-down cognitive control over emotional processing—potentially reflecting increased effort to regulate negative affect; (5) FPN receives increased reward network input with concurrent reduction in limbic system input, suggesting altered affective influence on cognitive control processes; and (6) LIN shows significant loss of input from salience networks, potentially disrupting appropriate emotional responses to salient stimuli. These circuit-level reception abnormalities align with core MDD symptoms including negative bias in

self-referential processing, emotional dysregulation, compensatory cognitive control, and impaired integration of salience and affective information. More analyses are provided in Appendix A.8.3.

## 5 CONCLUSION

We present NH-GCAT, a neurocircuitry-inspired hierarchical graph causal attention network that integrates temporal dynamics, hierarchical circuit encoding, and causal interactions for explainable MDD identification. NH-GCAT achieves state-of-the-art performance and provides interpretable insights into depression-related brain network alterations. Our findings underscore the value of embedding neuroscientific priors into deep learning frameworks to advance interpretable and clinically meaningful neuropsychiatric diagnosis.

## 6 REPRODUCIBILITY & ETHICS STATEMENT

**Reproducibility Statement.** To ensure our work is fully reproducible, we have made extensive efforts to document our methodology. We provide a detailed description of the dataset and preprocessing steps in Appendix A.1, and a comprehensive overview of our experimental setup, baseline implementations, and hyper-parameter settings in Appendices A.2 and A.4. Furthermore, the source code for our NH-GCAT model will be released publicly at `https://github.com/author/NH-GCAT` upon publication.

**Ethics Statement.** All authors have read and adhered to the ICLR Code of Ethics. Our research involves the secondary analysis of the REST-meta-MDD dataset, a publicly available resource containing fully anonymized human neuroimaging data. The original data collection was conducted under institutional review board (IRB) approval at each respective site, with written informed consent obtained from all participants. Our use of this pre-existing, de-identified data poses no additional privacy or security risks to the participants. The objective of this work is to develop computational methods that may positively contribute to the understanding and future clinical diagnosis of Major Depressive Disorder. There are no conflicts of interest or external sponsorships that influenced this work.

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

## A    TECHNICAL APPENDICES AND SUPPLEMENTARY MATERIAL

### A.1    DATASET DETAILS

This section provides detailed information about the REST-meta-MDD dataset used in our experiments, including data collection, preprocessing procedures, quality control, and demographic characteristics.

### A.1.1    REST-META-MDD DATASET

The REST-meta-MDD initiative (Yan et al., 2019) constitutes the largest multi-center neuroimaging repository for Major Depressive Disorder (MDD) research, accessible via the consortium's official platform[1]. This dataset aggregates resting-state fMRI (rs-fMRI) scans from 25 clinical centers across China, employing standardized rs-fMRI acquisition protocols to ensure cross-site consistency. A key methodological innovation lies in its federated preprocessing framework, where all participating sites implemented identical computational pipelines for spatial normalization and functional connectivity estimation prior to centralized analysis. This design explicitly addresses heterogeneity challenges in multi-site neuroimaging studies through protocol harmonization at both data acquisition and processing stages.

---

[1]Project portal: http://rfmri.org/REST-meta-MDD

**Sample Selection.** Following the protocols established in the original REST-meta-MDD publication (Yan et al., 2019), from the initial collection of 1,300 MDD patients and 1,128 healthy controls (HC), we selected 848 MDDs and 794 HCs from 17 sites for our analysis, yielding a preliminary dataset of 1,642 participants. All participants provided written informed consent, and the study protocols were approved by the local ethics committees of participating institutions.

**Quality Control.** Following REST-meta-MDD consortium guidelines (Chen et al., 2022), we excluded data from Site 4 due to duplication with Site 14 during quality control procedures. The final analytical cohort comprised 1,601 participants (830 MDD, 771 HC) distributed across 16 research sites after implementing standardized data cleaning protocols. This rigorous quality control procedures ensures data reliability.

Table 4: Demographic characteristics of participants across 16 sites in the REST-meta-MDD dataset.

| Site | Sample Size | | Age (Mean (SD)) | | Education (Mean (SD)) | | Sex (M/F) | |
|------|------|------|-------------|-------------|-------------|-------------|---------|---------|
| | MDD | HC | MDD | HC | MDD | HC | MDD | HC |
| Site 1 | 73 | 73 | 31.9 (8.1) | 31.7 (9.0) | 13.8 (3.0) | 15.2 (2.3) | 30/43 | 32/41 |
| Site 2 | 16 | 14 | 41.8 (11.5) | 45.6 (12.1) | 11.6 (4.5) | 10.0 (4.8) | 1/15 | 4/10 |
| Site 7 | 35 | 37 | 41.9 (11.7) | 38.2 (11.8) | 11.1 (4.0) | 14.9 (4.1) | 13/22 | 14/23 |
| Site 8 | 54 | 62 | 32.0 (9.6) | 31.1 (10.6) | 11.3 (3.2) | 13.1 (2.5) | 18/36 | 26/36 |
| Site 9 | 48 | 48 | 28.6 (8.7) | 28.6 (8.0) | 13.4 (2.9) | 15.9 (2.8) | 22/26 | 30/18 |
| Site 10 | 45 | 26 | 32.7 (10.8) | 32.7 (8.1) | 11.3 (3.1) | 12.8 (2.0) | 21/24 | 17/9 |
| Site 11 | 20 | 17 | 30.2 (9.3) | 31.4 (9.6) | 11.2 (3.0) | 15.6 (2.5) | 9/11 | 8/9 |
| Site 13 | 20 | 16 | 32.6 (8.6) | 34.4 (10.7) | 13.7 (2.2) | 13.2 (2.3) | 8/12 | 5/11 |
| Site 14 | 61 | 32 | 30.1 (7.0) | 29.6 (5.0) | 13.7 (3.3) | 14.6 (2.8) | 19/42 | 15/17 |
| Site 15 | 30 | 37 | 46.5 (12.6) | 39.8 (14.7) | 11.1 (3.8) | 13.1 (3.8) | 9/21 | 17/20 |
| Site 17 | 41 | 41 | 21.7 (3.0) | 20.6 (1.8) | 13.1 (1.5) | 13.8 (1.6) | 14/27 | 13/28 |
| Site 19 | 18 | 31 | 34.9 (11.4) | 35.2 (10.2) | 9.7 (3.1) | 9.9 (3.9) | 5/13 | 14/17 |
| Site 20 | 250 | 229 | 38.5 (11.9) | 39.6 (15.7) | 10.9 (3.4) | 13.0 (3.8) | 84/166 | 73/156 |
| Site 21 | 79 | 65 | 34.1 (12.1) | 36.5 (12.5) | 11.8 (2.7) | 13.0 (2.1) | 34/45 | 28/37 |
| Site 22 | 18 | 20 | 33.8 (9.8) | 24.4 (7.1) | 12.0 (3.0) | 13.3 (2.1) | 9/9 | 12/8 |
| Site 23 | 22 | 23 | 26.2 (7.4) | 33.0 (12.0) | 13.9 (3.2) | 14.3 (4.1) | 10/12 | 8/15 |
| **Total** | 830 | 771 | 34.4 (11.6) | 34.5 (13.2) | 11.9 (3.4) | 13.5 (3.4) | 306/524 | 316/455 |

### A.1.2 BRAIN PARCELLATION AND GRAPH CONSTRUCTION

**ROI Extraction.** Following preprocessing, we extracted regional BOLD time series from 116 anatomically defined regions using the Automated Anatomical Labeling (AAL) atlas (Tzourio-Mazoyer et al., 2002). This atlas was selected for its established validity in neuropsychiatric research and comprehensive coverage of cortical and subcortical structures implicated in depression pathophysiology. For each subject, we derived two complementary feature sets:

- **Temporal features:** 116 regional BOLD time series ($116 \times T$ matrix, where T represents the number of time points), capturing the dynamic neural activity patterns across the brain.

- **Multi-dimensional static features:** We implemented an overlapping sliding window approach (window length T=90, stride S=45) to extract: (1) a $116 \times 116$ functional connectivity matrix computed as Fisher z-transformed Pearson correlations between regional time series; (2) spectral characteristics including variance and low-frequency power (0.01-0.1 Hz) for each region, which captures neurobiologically relevant oscillations associated with resting-state networks; and (3) demographic variables including age, sex, and education level to account for potential confounding factors.

**Brain Graph Construction.** To construct brain graphs for our graph neural network approach, we employed a k-nearest neighbors (k=40) algorithm using the functional connectivity matrix as edge weights. This sparse graph construction approach preserves the strongest functional connections while reducing computational complexity and noise. We chose k=40 based on preliminary experiments indicating optimal performance while maintaining physiologically plausible network topology.

**Reference Graph Templates.** We constructed group-level reference graph templates by averaging functional connectivity matrices within diagnostic groups:

- MDD group-level template: Average connectivity pattern across all MDD subjects in the the training set.
- HC group-level template: Average connectivity pattern across all healthy controls in the training set.

These templates provided prior knowledge for our hierarchical circuit encoding scheme, enabling the model to learn connectivity patterns characteristic of each diagnostic group.

### A.1.3 CIRCUIT-SPECIFIC FEATURES

For neurocircuitry-informed analysis, we leveraged established neuroscientific knowledge to assign AAL regions to five depression-relevant neural circuits: Default Mode Network (DMN), Frontoparietal Network (FPN), Salience Network (SN), Limbic Network (LN), and Reward Network (RN). This assignment followed consensus mappings from multiple sources in the depression neuroimaging literature (Menon, 2011; Kaiser et al., 2015; Hamilton et al., 2011; Whitfield-Gabrieli & Ford, 2012).

### A.2 BASELINE COMPARISON METHODOLOGY

In Table 1 of the main paper, we present performance comparisons between NH-GCAT and various baseline methods. For transparency and to ensure fair comparison, we provide a detailed explanation of our comparison methodology below.

**Data Partitioning.** For model evaluation, we utilized two complementary strategies:

- 5-fold cross-validation: Data were randomly partitioned into 5 folds with stratification to maintain diagnostic class distribution.
- Leave-one-site-out cross-validation: Each site was sequentially held out as a test set, with the remaining 15 sites used for developing model.

This dual evaluation approach allowed us to assess both general performance and cross-site generalizability of our model.

**Comparison with Published State-of-the-Art Methods.** For specialized MDD classification methods, we report performance metrics as published in their respective papers. This approach is scientifically justified for several reasons:

1. **Common Dataset:** All compared methods were evaluated on the REST-meta-MDD dataset, the same dataset used in our study. As shown in Table 5, most studies used comparable sample sizes (approximately 1,600 subjects), with minor variations due to different quality control procedures.
2. **Standardized Brain Atlas:** The majority of compared methods (BrainIB, BPI-GNN, TEM, CI-GNN, LGMF-GNN, BrainNPT, MSSTAN) used the AAL atlas for brain parcellation, matching our approach and ensuring comparable region definitions.
3. **Similar Cross-validation Strategies:** Most methods employed either 5-fold or 10-fold cross-validation protocols, with LCCAF, GC-GAN, DSFGNN, and our approach specifically using 5-fold cross-validation.

**Addressing Methodological Variations.** While direct reimplementation of all baseline methods would be ideal, it presents several practical challenges:

1. **Implementation Complexity:** Many specialized methods involve complex architectures with numerous hyperparameters. Reimplementing these without access to original code could introduce unintentional modifications that affect performance.

Table 5: Detailed information about baseline methods and evaluation protocols. CV: Cross-Validation.

| Model | Validation Method | Atlas | Sample Size | Data Split |
|---|---|---|---|---|
| BrainIB | 10-fold CV | AAL | 1604 (848 MDD, 794 HC) | Not specified |
| MV-GNN | Leave-one-site-out CV | AAL | 1160 (597 MDD, 563 HC) | Not specified |
| GC-GAN | 5-fold CV | Harvard Oxford | 477 (249 MDD, 228 HC) | Not specified |
| DSFGNN | 5-fold CV | AAL | 1611 (832 MDD, 779 HC) | Not specified |
| BPI-GNN | Random split | AAL | 1604 (828 MDD, 776 HC) | 80%/10%/10% |
| TEM | 5-fold CV | AAL | 1611 (832 MDD, 779 HC) | Not specified |
| CI-GNN | Random split | AAL | 1604 (828 MDD, 776 HC) | 80%/10%/10% |
| LGMF-GNN | 10-fold CV | AAL | 1570 (814 MDD, 756 HC) | Not specified |
| BrainNPT | Random split | AAL | 2027 (1041 MDD, 986 HC) | 80%/10%/10% |
| MSSTAN | 10-fold CV | AAL | 667 (368 MDD, 299 HC) | Not specified |
| LCCAF | 5-fold CV | Craddock (CC) | 1601 (830 MDD, 771 HC) | Not specified |
| NH-GCAT (Ours) | 5-fold CV | AAL | 1601 (830 MDD, 771 HC) | Random stratified |

2. **Computational Constraints:** Training multiple deep learning models on large neuroimaging datasets requires substantial computational resources, particularly when hyperparameter optimization is necessary for fair comparison.

3. **Established Practice:** In neuroimaging machine learning research, comparing with published results on standardized datasets is an established practice, particularly when evaluating on large, publicly available datasets like REST-meta-MDD.

To mitigate potential concerns about comparison fairness, we took several additional steps:

1. **Our Implemented Baselines:** We implemented five general-purpose graph neural networks (GCN, GIN, GraphSAGE, GPS, GAT) ourselves using identical preprocessing, feature extraction, and evaluation protocols as our NH-GCAT model. This provides a controlled comparison with widely-used graph learning architectures.

2. **Consistent Evaluation Metrics:** We report the same set of evaluation metrics (AUC, accuracy, sensitivity, specificity, F1 score) as used in the original papers, enabling direct comparison.

3. **Multiple Evaluation Protocols:** We evaluated NH-GCAT using both 5-fold cross-validation (for comparison with most methods) and leave-one-site-out cross-validation (for comparison with recent state-of-the-art methods like BrainIB (Zheng et al., 2024c)), ensuring comprehensive benchmarking.

4. **Statistical Significance Testing:** We conducted rigorous statistical tests to verify that performance improvements are significant and not due to random variation.

This comprehensive approach to baseline comparison—combining published results from specialized methods with our own implementations of general architectures—provides a thorough and fair evaluation of NH-GCAT's performance within the current landscape of MDD classification methods.

## A.3 ARCHITECTURAL DESIGN RATIONALE AND COMPARISON WITH ALTERNATIVES

In this section, we elaborate on the rationale behind our key architectural design choices in NH-GCAT. Our overarching philosophy is to infuse neuroscientific domain knowledge as an architectural inductive bias, moving beyond purely data-driven approaches to create a model that is not only accurate but also mechanistically interpretable. We detail the specific motivations for our three core components: Residual Gated Fusion (RG-Fusion), Hierarchical Circuit Encoding (HC-Pooling) with ChildSumTreeLSTM, and the Variational Latent Causal Attention (VLCA) mechanism.

### A.3.1 RATIONALE FOR RESIDUAL GATED FUSION (RG-FUSION)

**Problem Formulation.** The pathophysiology of Major Depressive Disorder (MDD) manifests in both static and dynamic properties of brain networks. Static functional connectivity (FC) provides a

time-averaged summary of network topology, while temporal Blood Oxygenation Level Dependent (BOLD) signals capture dynamic, moment-to-moment neural fluctuations. Critically, depression is linked to altered low-frequency oscillations ($<0.1$ Hz), which are lost when relying solely on static FC matrices. Conventional Graph Neural Network (GNN) models for MDD classification often ignore this temporal dimension, leading to suboptimal feature extraction.

**Intuition and Design.**   The RG-Fusion module is explicitly designed to synergistically integrate these two complementary data modalities. It employs a dual-stream architecture:

1. A **temporal pathway** uses a Transformer Encoder to process the raw BOLD time series. The self-attention mechanism is particularly adept at capturing long-range temporal dependencies within the signal, which is crucial for modeling low-frequency oscillations.

2. A **static pathway** processes the FC matrix using standard graph convolutional layers to learn topological patterns.

The core innovation is the **adaptive gating mechanism**. Instead of simple concatenation, which would treat both feature streams equally, our gate learns to dynamically weight the importance of temporal versus static information for each brain region. This allows the model to selectively emphasize features most relevant to depression classification on a node-by-node basis. The residual connection ensures stable training and prevents the loss of critical information from the primary temporal pathway during fusion.

**Comparison to Alternatives.**

- **Static FC-based GNNs:** This is the most common approach but is fundamentally limited as it discards rich dynamic information contained in BOLD signals, particularly the depression-relevant oscillatory patterns.

- **Simple Feature Concatenation:** A naive concatenation of temporal and static features lacks the flexibility to adaptively prioritize information. Our learned gating mechanism provides a more principled fusion, allowing the model to determine the optimal balance between modalities, which can vary across brain regions and subjects.

### A.3.2    RATIONALE FOR HIERARCHICAL CIRCUIT ENCODING (HC-POOLING) AND CHILDSUMTREELSTM

**Problem Formulation.**   The human brain is not a flat, homogeneous graph; it possesses a well-established hierarchical organization. At a macroscopic level, brain regions form functional circuits (e.g., Default Mode Network (DMN), Salience Network (SN)), which collaboratively govern complex cognitive and emotional processes. Dysfunctions in MDD are often best understood at this circuit level. Standard GNN pooling mechanisms (e.g., global mean/max/sum pooling) are agnostic to this neurobiological reality, collapsing node features into a single vector and losing crucial circuit-specific information.

**Intuition and Design.**   The HC-Pooling module is designed to explicitly model the brain's multi-scale organization by aggregating regional node representations according to a predefined, neuro-scientifically validated circuit hierarchy. This transforms node-level embeddings into circuit-level embeddings, aligning the model's representations with the language of cognitive neuroscience.

**Justification for ChildSumTreeLSTM.**   To perform this hierarchical aggregation, we required an operator capable of processing information on a tree-structured hierarchy. The choice of Child-SumTreeLSTM over other alternatives was deliberate and principled:

1. **Alignment with Hierarchical Structure:** Unlike standard LSTMs or GRUs that operate on linear sequences, TreeLSTMs are specifically designed for tree-structured data. Our defined hierarchy, where brain regions (leaf nodes) are grouped into circuits (parent nodes), naturally forms a tree.

2. **Handling of Variable Branching Factors:** The "Child-Sum" variant is particularly suitable for our task. Neural circuits are not uniform in size; some contain many brain regions

(children), while others contain few. ChildSumTreeLSTM elegantly handles this variability by summing the hidden states of all child nodes before feeding them into the LSTM cell. This makes it a flexible and robust aggregator for real-world neuroanatomical structures.

**Comparison to Alternatives.**

- **Standard LSTMs/Sequence Models:** These are fundamentally incompatible as they cannot process the non-sequential, hierarchical relationships between brain regions within a circuit.

- **Generic GNN Layers for Pooling:** One could stack more GNN layers to achieve a global representation, but this does not explicitly create distinct, interpretable embeddings for each predefined circuit. Our approach guarantees that the resulting vectors correspond to the DMN, FPN, etc.

- **Other Hierarchical Pooling Methods (e.g., DiffPool):** Methods like DiffPool learn a hierarchical structure in a purely data-driven manner. While powerful, our objective was to *leverage* established neuroscientific knowledge as a strong prior. By using a predefined hierarchy and a structure-aware aggregator like ChildSumTreeLSTM, we ensure that the model's internal organization is neurobiologically meaningful and its subsequent analyses are directly interpretable in the context of existing depression literature.

### A.3.3 RATIONALE FOR VARIATIONAL LATENT CAUSAL ATTENTION (VLCA)

**Problem Formulation.** For a model to be truly explainable, it must move beyond identifying *correlations* to inferring *directed influence*. We need to understand how dysfunction in one neural circuit might causally impact others. Standard attention mechanisms in GNNs identify which nodes or features are important for a prediction but do not typically model directionality or provide a framework for causal reasoning.

**Intuition and Design.** The VLCA mechanism is designed to model the directed information flow between the high-level neural circuits derived from HC-Pooling. It achieves this through two key innovations:

1. **Variational Framework:** By encoding the learned circuit interactions into a continuous probabilistic latent space, the model learns a robust and smooth representation of inter-circuit dynamics, capturing uncertainty in these complex biological systems.

2. **Counterfactual Reasoning:** The core of the causal inference lies in comparing the model's output under two conditions: (a) using the learned, attention-weighted interactions (`real`), and (b) using a counterfactual scenario where these interactions are removed (i.e., attention is replaced with self-attention only via an identity matrix). The difference in outcomes allows us to estimate the *causal effect* of inter-circuit communication on the classification of depression. This is integrated directly into the training objective.

**Comparison to Alternatives.**

- **Standard Graph Attention (GAT):** GAT computes scalar attention weights that indicate feature importance. It does not inherently model the directional flow of information between high-level conceptual units (our circuits) or provide a mechanism to quantify the causal impact of these interactions.

- **Post-hoc Explainability Methods (e.g., GNNExplainer, Integrated Gradients):** These methods analyze a trained model to find important features or subgraphs. While useful, they are separate from the learning process. VLCA integrates causal reasoning directly into the model's architecture and objective function. This encourages the model to learn representations that are inherently causal and interpretable from the outset, rather than attempting to explain a black box after the fact. This architecture-constrained approach generally leads to more robust and faithful explanations.

### A.4 Implementation Details

This section provides comprehensive details about the architecture specifications, hyperparameter settings, and training procedures of NH-GCAT to facilitate reproducibility.

#### A.4.1 Architecture Specifications

**Feature and Node Attention.** Two-stage attention with feature-wise attention (single-head, temperature=0.1) followed by node-wise attention (single-head, temperature=0.1).

**Variational Encoder.** 2-layer MLP (hidden dims= 32, 16) for mean and log-variance estimation.

**Classifier.** The final classification is performed by:

- **Circuit Integration:** Concatenation of circuit-level embeddings followed by a 2-layer MLP (hidden dims=128, 64) with dropout=0.5.
- **Output Layer:** Linear layer with 2-dimensional output and softmax activation.

**Network Dimensions.** The NH-GCAT model maintains consistent hidden dimensions across its components, with the primary embedding dimension set to 128. Specific dimensional configurations for each module are:

- **RG-Fusion**: The transformer encoder for BOLD signal processing uses 4 attention heads with a hidden dimension of 128. The subsequent graph encoding layers (SAGEConv and GATConv) both produce 64-dimensional outputs that are concatenated to form 128-dimensional node representations.
- **HC-Pooling**: Each circuit-specific hierarchical encoding maintains 128-dimensional representations across all three hierarchical levels. The ChildSumTreeLSTM uses 128-dimensional hidden and cell states.
- **VLCA**: The causal attention mechanism employs single-head attention with a 64-dimensional output. The variational encoder projects these into a latent space with dimension 32.

**Activation Functions.** We employ Leaky ReLU (negative slope = 0.2) for all graph convolutional operations and MLPs within the RG-Fusion module. The gating mechanisms use sigmoid activations, while the ChildSumTreeLSTM follows the standard LSTM activation pattern with tanh and sigmoid functions.

**Normalization and Regularization.** Layer normalization is applied after each transformer encoder block. We employ dropout (rate = 0.2) after each convolutional operation and within the attention mechanisms. For the probabilistic components, we use a KL divergence regularization term with dynamic weighting.

**Parameters and Network Size.** Our final NH-GCAT model has approximately 2.1 million trainable parameters.

#### A.4.2 Hyperparameter Settings

Table 6 summarizes the key hyperparameters used in our experiments.

#### A.4.3 Training Procedure

We employed the Adam optimizer with an initial learning rate of 1e-3 and weight decay of 0.1. To stabilize training, we implemented gradient clipping with a maximum norm of 1.0. For regularization terms, we used dynamic weight scheduling where $\lambda_{kl}$ increases linearly from 0 to 0.1 during the first 20 epochs, and $\lambda_{mse}$ follows a cosine schedule between 0.2 and 1.0 over the course of training.

Table 6: Hyperparameter settings for NH-GCAT.

| Hyperparameter | Value |
|---|---|
| Learning rate | 1e-3 |
| Weight decay | 0.1 |
| Batch size | 32 |
| Training epochs | 300 |
| Early stopping patience | 20 |
| Dropout rate | 0.5 |
| Gradient clipping norm | 1.0 |
| $\lambda_{kl}$ (KL divergence weight) | $0.0 \rightarrow 0.1$ (linear schedule) |
| $\lambda_{VLCA}$ (VLCA loss weight) | 1.0 |
| $\lambda_{mse}$ (MSE loss weight) | $0.2 \rightarrow 1.0$ (cosine schedule) |
| Temperature for Gumbel-Softmax | $1.0 \rightarrow 0.5$ (exponential decay) |
| Graph construction $k$ (KNN) | 40 |

Training proceeded for a maximum of 300 epochs with early stopping (patience = 20) based on validation performance. The best-performing checkpoint was selected for final evaluation. During training, we dynamically balanced loss terms by applying adaptive weight reduction when specific loss components exceeded predefined thresholds.

For data augmentation, we employed random edge dropout (10%) during training to enhance robustness. The model was trained using a 5-fold stratified cross-validation procedure, ensuring consistent class distribution across folds. For leave-one-site-out validation, we trained on data from 15 sites and tested on the held-out site, repeating this procedure for all 16 sites.

### A.4.4 IMPLEMENTATION ENVIRONMENT

All experiments were implemented using PyTorch 2.5.1 and PyTorch-Geometric 2.6.1. For circuit-specific operations, we developed custom extensions to PyTorch-Geometric to support hierarchical graph operations. Our custom implementation of the ChildSumTreeLSTM was based on the DGL (Deep Graph Library) framework but optimized for our specific hierarchical circuit structure.

### A.4.5 CODE AVAILABILITY

The implementation code for NH-GCAT will be made publicly available at `https://github.com/author/NH-GCAT` upon publication.

### A.5 EXTENDED PERFORMANCE AND CLINICAL UTILITY ANALYSIS

To provide a more comprehensive and nuanced evaluation of the proposed NH-GCAT framework, this section extends the performance analysis presented in the main paper. We supplement the primary classification metrics with detailed visualizations of the Receiver Operating Characteristic (ROC) curve, the Precision-Recall (PR) curve, and a Decision Curve Analysis (DCA). These analyses, based on the 5-fold cross-validation results, offer deeper insights into the model's discriminative ability, its performance on the positive class (MDD), and its potential clinical utility.

### A.5.1 RECEIVER OPERATING CHARACTERISTIC (ROC) ANALYSIS

The ROC curve, shown in Figure 4a, illustrates the trade-off between the true positive rate (Sensitivity) and the false positive rate (1 - Specificity) at various classification thresholds. A model with strong discriminative capability will have a curve that bows towards the top-left corner.

Our NH-GCAT model achieves a mean Area Under the Curve (AUC) of $0.786 \pm 0.017$ across the five folds. The consistency across folds, indicated by the narrow shaded region representing the standard deviation, highlights the model's stability. This result reinforces the findings from Table 1 in the main paper, confirming that NH-GCAT is highly effective at distinguishing between individuals with MDD and healthy controls.

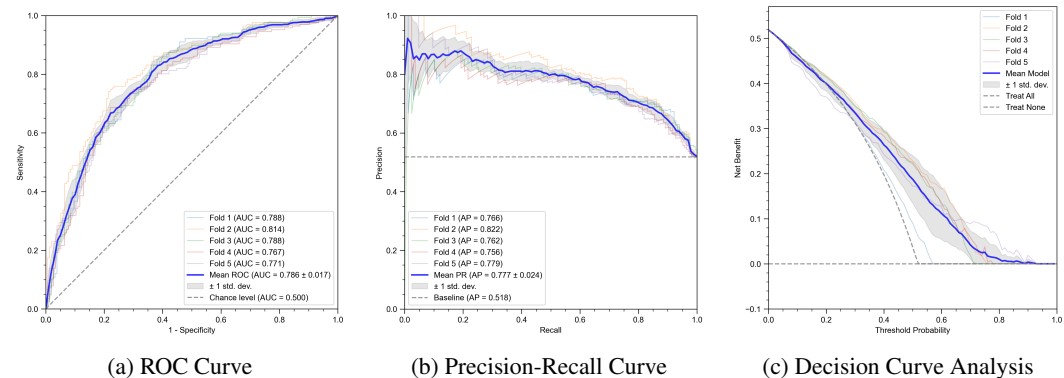

(a) ROC Curve      (b) Precision-Recall Curve      (c) Decision Curve Analysis

Figure 4: Comprehensive performance evaluation of NH-GCAT across 5 cross-validation folds.

### A.5.2 PRECISION-RECALL (PR) ANALYSIS

While the ROC curve provides a general view of discriminative performance, the Precision-Recall (PR) curve (Figure 4b) is particularly informative for evaluating a model's ability to correctly identify the positive class, which in our case are the MDD subjects. This is clinically crucial, as failing to identify a patient (a false negative) can have significant consequences.

NH-GCAT achieves a mean Average Precision (AP) of $0.777 \pm 0.024$. This is substantially higher than the baseline AP of 0.518, which corresponds to the proportion of positive samples in the dataset. The consistently high precision across a wide range of recall values indicates that when the model identifies a subject as having MDD, it is likely to be correct, and it can do so without missing a large number of actual MDD cases. This demonstrates the model's reliability for screening or diagnostic support applications.

### A.5.3 DECISION CURVE ANALYSIS (DCA) FOR CLINICAL UTILITY

Beyond standard statistical metrics, it is vital to assess whether a predictive model offers tangible benefits in a clinical setting. Decision Curve Analysis (DCA) is a method for evaluating the clinical utility of a model by quantifying its net benefit across a range of risk thresholds for intervention. The net benefit is calculated by balancing the benefits of true positives against the harms of false positives.

Figure 4c presents the DCA for our NH-GCAT model. The x-axis represents the threshold probability, which is the risk threshold at which a clinician (or a policy) would decide to intervene (e.g., recommend further testing or treatment). The y-axis shows the net benefit. A model is considered clinically useful if its net benefit is higher than the two default strategies: "Treat None" (net benefit is always zero) and "Treat All".

The curve for NH-GCAT (Mean Model) demonstrates a positive net benefit across a wide and clinically relevant range of threshold probabilities, approximately from **0.10 to 0.75**. This means that using the NH-GCAT model to guide clinical decisions would lead to better outcomes than either treating all patients or treating none of them within this wide decision-making range. This analysis provides strong evidence that our model's predictions are not just statistically significant but also translate into practical clinical value, justifying the use of a sophisticated, interpretable model for this high-stakes task.

### A.6 LEAVE-ONE-SITE-OUT CROSS-VALIDATION RESULTS

This section provides a detailed analysis of our leave-one-site-out cross-validation results, complementing the summary presented in Section 4.2 (Performance Comparison). As noted in the main paper, NH-GCAT achieves the highest accuracy on 8 out of 16 sites (50%) with an weighted-average accuracy of 73.3% across all sites, demonstrating a +4.1% improvement over the best competing methods.

Table 7 presents the comprehensive performance metrics for our NH-GCAT model across all evaluation sites in the REST-meta-MDD dataset. The results highlight our model's ability to generalize across heterogeneous data collection sites with varying sample sizes and demographic characteristics.

Table 7: Detailed leave-one-site-out cross-validation performance metrics of NH-GCAT across 16 sites from the REST-meta-MDD dataset. For each metric, the best value is shown in bold and the second-best value is underlined. The final row shows the sample-size weighted average.

| Site (MDD/HC) | Sensitivity | Specificity | Accuracy | F1 Score | AUC |
|---|---|---|---|---|---|
| Site 1 (73/73) | 61.6 | 74.0 | 67.8 | 65.7 | 70.2 |
| Site 2 (16/14) | **100.0** | 57.1 | 80.0 | 84.2 | 85.3 |
| Site 3 (35/37) | 71.4 | 73.0 | 72.2 | 71.4 | 77.5 |
| Site 4 (54/62) | 72.2 | 72.6 | 72.4 | 70.9 | 76.3 |
| Site 5 (48/48) | 81.2 | 85.4 | 83.3 | 83.0 | 88.5 |
| Site 6 (45/26) | 68.9 | 76.9 | 71.8 | 75.6 | 69.4 |
| Site 7 (20/17) | 90.0 | 82.4 | **86.5** | **87.8** | **93.5** |
| Site 8 (20/16) | 55.0 | 87.5 | 69.4 | 66.7 | 63.4 |
| Site 9 (61/32) | 77.0 | 62.5 | 72.0 | 78.3 | 67.4 |
| Site 10 (30/37) | 76.7 | 73.0 | 74.6 | 73.0 | 78.1 |
| Site 11 (41/41) | 75.6 | 63.4 | 69.5 | 71.3 | 73.4 |
| Site 12 (18/31) | 50.0 | **100.0** | 81.6 | 66.7 | 79.4 |
| Site 13 (250/229) | 72.0 | 74.7 | 73.3 | 73.8 | 78.7 |
| Site 14 (79/65) | 60.8 | 78.5 | 68.8 | 68.1 | 71.0 |
| Site 15 (18/20) | 88.9 | 60.0 | 73.7 | 76.2 | 76.7 |
| Site 16 (22/23) | 90.9 | 65.2 | 77.8 | 80.0 | 82.4 |
| **Unweighted Average** | 74.5 (13.7) | 74.1 (11.2) | 74.7 (5.6) | 74.5 (6.7) | 77.0 (7.9) |
| **Weighted Average** | 71.9 (9.5) | 74.4 (7.9) | 73.3 (4.4) | 73.3 (5.2) | 76.4 (6.1) |

Several key observations can be drawn from these results:

1. **Robustness across sample sizes:** NH-GCAT performs well on both large sites (e.g., Site 13 with 250 MDD/229 HC) and small sites (e.g., Site 7 with 20 MDD/17 HC), demonstrating its ability to learn meaningful representations regardless of sample size. This is particularly evident in Site 7, where our model achieves the highest accuracy (86.5%) and AUC (93.5%) despite the limited sample.

2. **Performance on balanced vs. imbalanced sites:** The model maintains strong performance on both balanced sites (e.g., Site 5 with 48 MDD/48 HC) and imbalanced sites (e.g., Site 12 with 18 MDD/31 HC), indicating robustness to class distribution variations.

3. **Consistent sensitivity:** In alignment with our findings in the main paper, NH-GCAT demonstrates high sensitivity (74.5% average) across sites, which is clinically valuable for depression screening applications where identifying potential MDD cases is prioritized.

4. **Significant improvements on challenging sites:** As noted in Section 4.2, our model shows substantial improvements on sites where previous methods struggled, particularly on larger sites like Site 13 (+10.1% improvement) and Site 7 (+10.8% improvement).

These detailed results further validate the effectiveness of our neurocircuitry-informed approach. By incorporating domain knowledge about depression-related neural circuits through our hierarchical circuit encoding scheme, NH-GCAT can better capture the complex patterns of functional dysregulation characteristic of MDD across diverse clinical populations. The model's strong performance in this rigorous cross-validation setting demonstrates its potential for real-world clinical applications where generalization across heterogeneous data sources is essential.

## A.7 EXTENDED ABLATION STUDIES

To thoroughly evaluate the contribution of each component in NH-GCAT, we conducted extensive ablation studies beyond those presented in the main paper. Table 8 provides a comprehensive comparison of different architectural variants across all evaluation metrics.

Table 8: Extended ablation study showing the contribution of each component and design choice in the NH-GCAT framework. The best results are marked in bold and the standard deviations are in parentheses. *Statistically significant improvement over GAT-Baseline ($p < 0.05$, Wilcoxon signed-rank test).

| Model Variant | AUC (%) | ACC (%) | SEN (%) | SPE (%) | F1 (%) |
|---|---|---|---|---|---|
| GAT-Baseline | 71.5 (3.2) | 67.7 (2.7) | **77.5 (9.1)** | 57.2 (9.4) | 71.2 (3.3) |
| + MLP-Fusion | 72.8 (3.2) | 68.4 (3.2) | 72.7 (9.9) | 63.8 (10.2) | 70.2 (4.3) |
| + Transformer-Fusion | 73.6 (1.5) | 69.7 (2.0) | 71.7 (6.8) | 67.6 (4.4) | 70.9 (3.2) |
| + RG-Fusion | 74.8 (2.3) | 70.2 (1.7) | 69.9 (10.9) | 70.6 (9.9) | 70.5 (4.3) |
| *VLCA variants (with RG-Fusion)* | | | | | |
| + Standard attention | 72.4 (3.4) | 68.3 (2.5) | 71.7 (7.2) | 64.7 (6.3) | 70.0 (3.4) |
| + Deterministic causal | 74.0 (3.3) | 70.1 (3.1) | 70.1 (11.9) | 70.2 (8.9) | 70.5 (5.1) |
| + Variational (no causal) | 71.9 (3.1) | 67.4 (1.9) | 65.4 (10.0) | 69.5 (9.8) | 67.2 (3.8) |
| + VLCA (full) | 75.9 (2.0) | 72.0 (2.0) | 75.4 (5.4) | 68.2 (6.5) | 73.6 (2.1) |
| *HC-Pooling variants (with RG-Fusion + VLCA)* | | | | | |
| + 1-layer hierarchy | 74.9 (2.2) | 69.6 (1.5) | 72.4 (2.7) | 66.5 (4.5) | 71.2 (1.2) |
| + 2-layer hierarchy | 75.4 (1.8) | 70.8 (1.1) | 74.8 (4.3) | 66.5 (4.9) | 72.6 (1.5) |
| + 3-layer hierarchy | **78.5 (1.7)*** | **73.8 (1.4)*** | 76.4 (5.8) | **71.0 (6.6)*** | **75.0 (1.8)*** |
| + 4-layer hierarchy | 76.1 (1.8) | 72.5 (1.5) | 74.1 (6.4) | 70.7 (4.7) | 73.5 (2.6) |

### A.7.1 ANALYSIS OF VLCA VARIANTS

Building upon the RG-Fusion module, we evaluated four variants of the causal attention mechanism to assess the contribution of both variational encoding and causal modeling:

- **Standard attention:** Multi-head attention without variational encoding or causal modeling.
- **Deterministic causal:** Causal attention without variational encoding.
- **Variational (no causal):** Variational encoding without causal modeling.
- **VLCA (full):** Our complete variational latent causal attention mechanism.

The full VLCA model consistently outperforms simpler attention mechanisms, with notable improvements in AUC (+3.5% over standard attention) and accuracy (+3.7% over standard attention). Interestingly, the deterministic causal variant achieves the highest specificity (70.2%), while the full VLCA model provides the best balance across all metrics. This confirms the importance of modeling both uncertainty and directionality in relationships between neural circuits for accurate depression classification.

### A.7.2 ANALYSIS OF HC-POOLING VARIANTS

With the RG-Fusion and VLCA components in place, we compared four different depths of hierarchical circuit encoding to evaluate the optimal architecture for capturing depression-related neuro-circuitry:

- **1-layer hierarchy:** A shallow hierarchical structure with limited capacity to model complex circuit interactions.
- **2-layer hierarchy:** A two-level hierarchical organization that captures basic circuit-level relationships.

- **3-layer hierarchy:** Our complete three-level differentiable hierarchical pooling that aligns with established neurocircuitry principles.

- **4-layer hierarchy:** A deeper hierarchical structure that may introduce unnecessary complexity.

Results demonstrate that the 3-layer HC-Pooling architecture achieves the best overall performance, with AUC (78.5%), accuracy (73.8%), and F1-score (75.0%) all reaching peak values. This confirms our hypothesis that a three-level hierarchy best captures the organizational principles of depression-related neural circuits. While the 4-layer variant achieves comparable specificity (70.7%) to the 3-layer model (71.0%), it shows reduced performance in other critical metrics including AUC (-2.4%), accuracy (-1.3%), and F1-score (-1.5%), suggesting potential overfitting with excessive hierarchical complexity. The progressive improvement from 1-layer to 3-layer hierarchy demonstrates clear benefits of increased hierarchical depth, with AUC improving from 74.9% to 78.5%, while the performance degradation at 4 layers indicates an optimal complexity threshold.

The ablation studies collectively demonstrate that each component of NH-GCAT contributes significantly to its overall performance. The progressive improvements from the baseline GAT model to the full NH-GCAT architecture highlight the value of our neuroscience-inspired approach. The RG-Fusion module substantially enhances specificity (+13.4%), addressing the baseline's primary weakness, while the VLCA mechanism with full variational causal modeling outperforms simpler attention variants, improving AUC by +1.1% and F1-score by +3.1% over RG-Fusion alone. The 3-layer HC-Pooling architecture provides the optimal hierarchical structure for modeling depression neurocircuitry, contributing final improvements of +2.6% in AUC and +1.4% in F1-score. These findings support our approach to integrating neuroscience domain knowledge with deep learning for MDD classification, with each design choice validated through systematic ablation analysis.

### A.7.3 ANALYSIS OF RG-FUSION VARIANTS

To investigate whether the performance gain of RG-Fusion stems merely from the inclusion of temporal data or specifically from our architectural design, we implemented two intermediate fusion variants for comparison (Table 8):

- **MLP-Fusion:** A baseline approach where temporal features (processed by a MLP) and static FC features are naively concatenated and fused via a Multilayer Perceptron.

- **Transformer-Fusion:** A stronger baseline using our Transformer Encoder to extract temporal dynamics, but fusing them with static features via simple summation/concatenation without the adaptive gating mechanism.

**Analysis of Results:** As shown in Table 8, while incorporating temporal information generally improves AUC compared to the static GAT-Baseline, the *method* of fusion is critical:

**Limitations of Naive Fusion:** `MLP-Fusion` offers only marginal improvements in AUC (+1.3%) and Accuracy (+0.7%). It fails to fully correct the model's bias, as evidenced by the relatively low Specificity (63.8%).

**Impact of Advanced Feature Extraction:** `Transformer-Fusion` outperforms `MLP-Fusion` (AUC 73.6% vs. 72.8%), confirming that the self-attention mechanism captures superior temporal representations of BOLD signals compared to simpler methods.

**Necessity of Adaptive Gating:** Our proposed **RG-Fusion** achieves the best overall performance (AUC 74.8%, ACC 70.2%). Most notably, it dramatically improves **Specificity** to 70.6% (a +13.4% gain over GAT-Baseline and +3.0% over Transformer-Fusion).

**Conclusion:** The results suggest that depression-related patterns are not uniformly distributed across static and dynamic modalities. The static GAT-Baseline tends to over-diagnose (High Sensitivity, Low Specificity). By employing the residual gating mechanism, RG-Fusion dynamically weighs the contribution of temporal vs. static features for each brain region. This effectively filters out false positives, leading to a much more balanced and clinically reliable diagnostic model.

## A.8 DETAILED INTERPRETABILITY ANALYSIS

This section provides an in-depth analysis of the interpretable components of NH-GCAT, examining how each module contributes to model explainability and offers neurobiologically meaningful insights into MDD pathophysiology.

### A.8.1 FREQUENCY-SPECIFIC NEURAL DYNAMICS ANALYSIS

To validate our RG-Fusion module's ability to capture depression-relevant neural oscillations, we conducted a frequency-specific analysis by separately feeding low-frequency (0.01-0.08 Hz) and high-frequency (0.1-0.25 Hz) BOLD signals into the trained model.

**Experimental Setup.** We filtered the original BOLD signals into two frequency bands using a bandpass filter implemented in the preprocessing pipeline:

- Low-frequency band (0.01-0.08 Hz): Known to contain depression-relevant neural oscillations (Calhoun et al., 2014)
- High-frequency band (0.1-0.25 Hz): Typically considered to contain physiological noise and artifacts

For each frequency band, we performed 5-fold cross-validation using identical train/test splits and model parameters as in our main experiments. We then compared the classification performance (AUC) between the two frequency bands.

**Results.** Figure 3(a) illustrates the performance comparison between low-frequency and high-frequency inputs. The model achieved significantly higher AUC with low-frequency inputs (mean=0.742, SD=0.019) compared to high-frequency inputs (mean=0.679, SD=0.032). A paired t-test confirmed the statistical significance of this difference ($p = 0.0037$).

Table 9: AUC values for low-frequency and high-frequency BOLD inputs across 5-fold cross-validation.

| Fold | Low-frequency AUC | High-frequency AUC |
|------|-------------------|--------------------|
| Fold 1 | 0.7549 | 0.7262 |
| Fold 2 | 0.7694 | 0.6894 |
| Fold 3 | 0.7187 | 0.6447 |
| Fold 4 | 0.7243 | 0.6425 |
| Fold 5 | 0.7409 | 0.6917 |
| **Mean (SD)** | **0.742 (0.019)** | **0.679 (0.032)** |

**Neurobiological Interpretation.** These findings confirm that our RG-Fusion module effectively captures depression-relevant neural oscillations predominantly manifested in low-frequency BOLD dynamics. This aligns with previous research indicating that depression-related functional connectivity alterations are most pronounced in the low-frequency band (Calhoun et al., 2014; Ding, 2025). The model's ability to leverage these frequency-specific patterns contributes to its superior classification performance compared to models that rely solely on static functional connectivity.

### A.8.2 HIERARCHICAL CIRCUIT ORGANIZATION ANALYSIS

We analyzed directional differences in hierarchical layer distributions between MDD and HC groups across depression-related neural circuits, as shown in Figure 5. The mapping between neural circuits and AAL regions is as follows: 1) DMN: Angular_L, Angular_R, Cingulum_Post_L, Cingulum_Post_R, Frontal_Sup_Medical_L, Frontal_Sup_Medical_R, Precuneus_L, Precuneus_R; 2) FPN: Frontal_Inf_Oper_L, Frontal_Inf_Oper_R, Frontal_Mid_L, Frontal_Mid_R, Parietal_Inf_L, Parietal_Inf_R; 3) LIN: Amygdala_L, Amygdala_R, Hippocampus_L, HIppocampus_R, ParaHippocampal_L, ParaHippocampal_R; 4) RN: Caudate_L, Caudate_R, Frontal_Mid_Orb_L, Frontal_Mid_Orb_R, Pallidum_L, Pallidum_R, Putamen_L, Putamen_R; 5) SN: Cingulum_Ant_L, Cingulum_Ant_R, Insula_L, Insula_R.

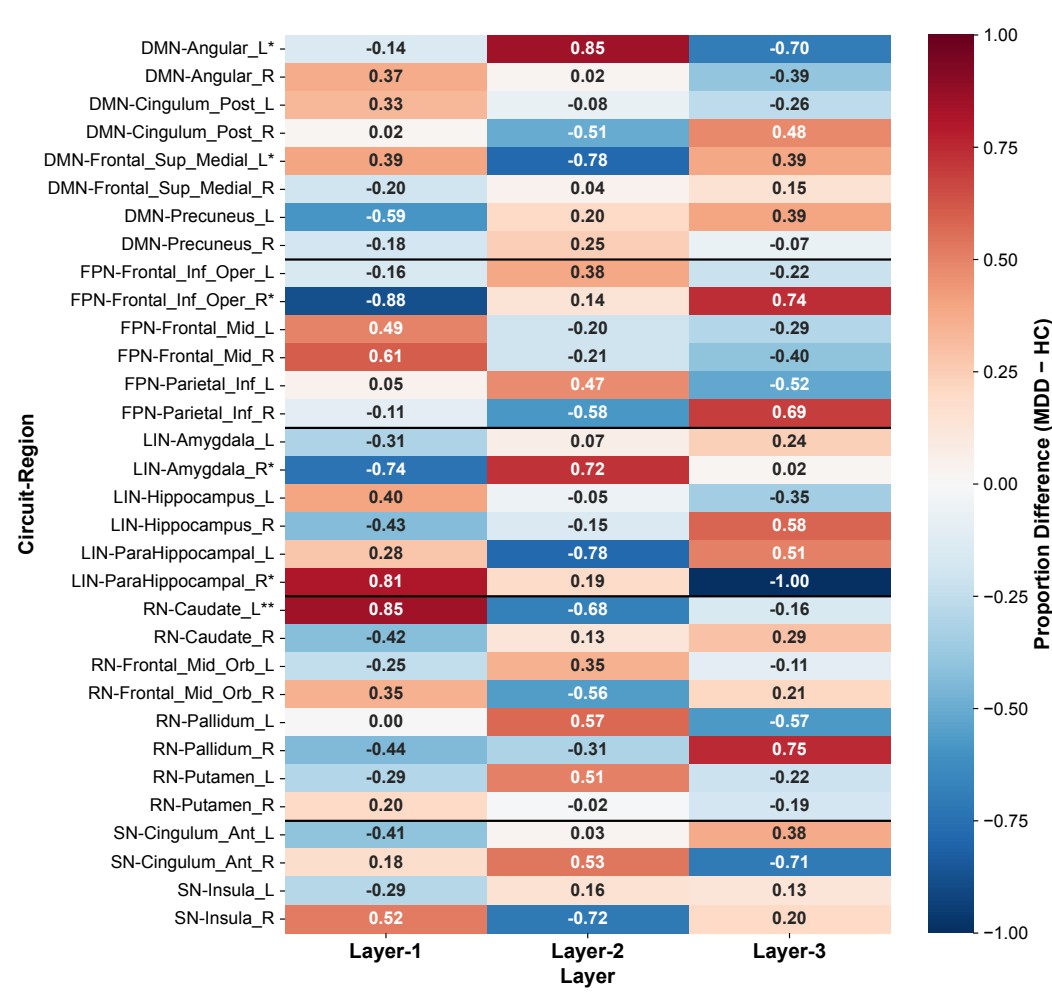

*Positive values (red): MDD > HC, Negative values (blue): HC > MDD*

Figure 5: Directional differences in hierarchical layer distributions between MDD and HC groups. Positive values (red) indicate higher proportions in MDD, negative values (blue) indicate higher proportions in HC. $*p < 0.05$, $**p < 0.01$.

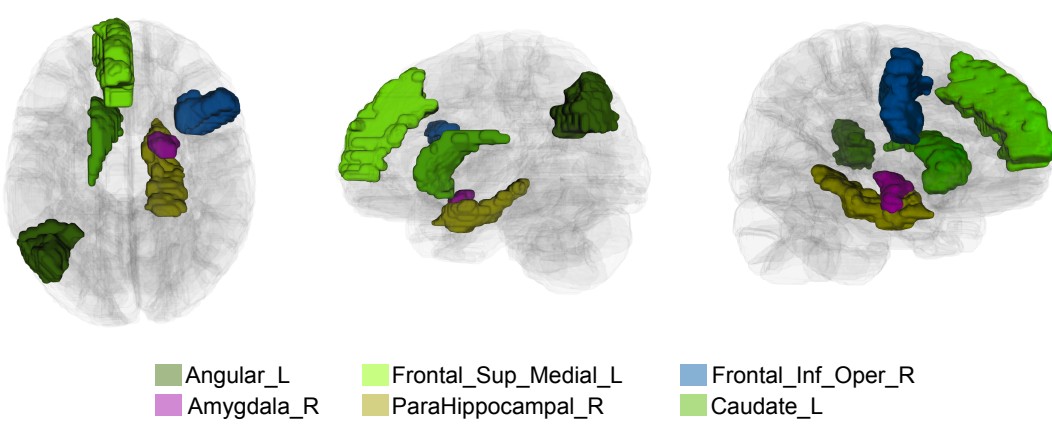

Figure 6: Spatial locations of the significant brain regions in the AAL atlas.

**Analysis Method.** For each subject, our HC-Pooling module assigned brain regions to three hierarchical levels (Layer-1: high-level integration, Layer-2: intermediate processing, Layer-3: primary processing). For each brain region, we calculated the proportion of subjects in each diagnostic group (MDD and HC) that assigned the region to each layer. We then computed the directional difference between these proportions (MDD - HC) and normalized these differences to the range [-1, 1] by dividing by the maximum absolute difference across all regions and layers. This normalization preserves the directionality of effects while enabling direct comparison across regions. Positive values indicate higher proportions in MDD, while negative values indicate higher proportions in HC. Statistical significance was assessed using Chi-square tests of independence.

**Results.** Our analysis revealed significant between-group differences in hierarchical organization across multiple circuits, with six regions showing statistically significant alterations (Figure 5), and their spatial locations in AAL atlas are shown in Figure 6.

**Circuit-specific Interpretations.** The directional differences reveal distinct patterns of hierarchical reorganization in MDD:

1. **Default Mode Network (DMN):** MDD exhibits a bidirectional reorganization with increased Layer-1 representation in Frontal_Sup_Medial_L (0.39, $p < 0.05$) but decreased Layer-1 in Angular_L (-0.14, $p < 0.05$). This suggests a functional imbalance within the DMN, with hyperactivity in medial prefrontal regions (associated with self-referential processing) and altered integration in parietal nodes. This pattern aligns with the pathological rumination and altered self-focus characteristic of depression.

2. **Frontoparietal Network (FPN):** Frontal_Inf_Oper_R shows substantially decreased Layer-1 representation (-0.88, $p < 0.05$) and increased Layer-3 representation (0.74) in MDD, indicating a significant reduction in high-level integration of this key cognitive control region. This supports the executive dysfunction hypothesis of depression, where impaired top-down control contributes to negative cognitive biases and difficulty disengaging from negative stimuli.

3. **Limbic Network (LIN):** We observed opposing patterns in limbic regions: ParaHippocampal_R showed increased Layer-1 representation (0.81, $p < 0.05$) while Amygdala_R showed decreased Layer-1 (-0.74, $p < 0.05$) and increased Layer-2 (0.72) representation in MDD. This suggests a reorganization of emotional processing circuits, with altered integration between memory-related (parahippocampal) and emotion-generating (amygdala) regions, consistent with emotional dysregulation in depression.

4. **Reward Network (RN):** Caudate_L showed the strongest effect, with significantly higher Layer-1 representation in MDD (0.85, $p < 0.01$) and lower Layer-2 (-0.68). This substantial reorganization of a key reward processing region may reflect compensatory mechanisms for anhedonia, with increased high-level integration potentially serving to counteract reward deficits.

These findings demonstrate how our HC-Pooling module captures clinically meaningful alterations in circuit hierarchy that align with established neurobiological models of depression. The directional nature of these differences provides novel insights into the specific reorganization patterns across hierarchical layers that may contribute to depression pathophysiology.

A.8.3 CAUSAL INTER-CIRCUIT INTERACTION ANALYSIS

We leveraged our VLCA mechanism to examine directed information flow among neural circuits, revealing distinct patterns of information reception in MDD versus HC groups.

**Analysis Method.** For each subject, we extracted the attention weights from the VLCA module, representing the strength of directed connections between circuits. To focus on the most significant connections while reducing noise, we first computed group averages for MDD and HC subjects, then applied a graph pruning technique that retained only the top-2 strongest outgoing connections (excluding self-connections) for each circuit. Finally, we normalized the weights across both groups to facilitate between-group comparison. The normalized weights were visualized as chord diagrams (Figure 3(c-d)).

**Results.** Quantitative analysis of the attention weights revealed several key differences in circuit-level information reception between MDD and HC groups, as detailed in Table 10.

Table 10: Circuit-level attention weights for MDD and HC groups after top-2 connection pruning.

| Source Circuit | Target Circuit | HC Weight | MDD Weight |
|---|---|---|---|
| DMN | SN | 0.846 | 0.652 |
| DMN | LIN | 0.685 | 0.614 |
| FPN | DMN | 0.361 | 0.006 |
| FPN | LIN | 0.000 | 0.502 |
| FPN | RN | 0.000 | 0.082 |
| SN | DMN | 0.000 | 0.000 |
| SN | FPN | 0.000 | 0.000 |
| LIN | DMN | 0.113 | 0.000 |
| LIN | FPN | 0.586 | 0.174 |
| LIN | SN | 0.479 | 0.000 |
| LIN | RN | 0.301 | 0.150 |
| RN | DMN | 0.000 | 0.476 |
| RN | FPN | 0.330 | 0.475 |
| RN | SN | 0.000 | 1.000 |
| RN | LIN | 0.398 | 0.000 |

**Neurobiological Interpretation.** Our analysis revealed six key alterations in circuit-level information reception in MDD:

1. **Altered DMN Information Reception:** In MDD, DMN receives significantly reduced input from frontoparietal networks (FPN→DMN: 0.361 in HC vs. 0.006 in MDD) and limbic networks (LIN→DMN: 0.113 in HC vs. 0.000 in MDD), while receiving novel input from reward networks (RN→DMN: 0.000 in HC vs. 0.476 in MDD). This reconfiguration suggests impaired cognitive and emotional regulation of self-referential processing, with abnormal integration of reward signals—potentially underlying negative self-focused rumination characteristic of depression.

2. **Reduced Salience Network Modulation:** SN receives diminished regulatory input from DMN (DMN→SN: 0.846 in HC vs. 0.652 in MDD) and complete loss of emotional input from limbic networks (LIN→SN: 0.479 in HC vs. 0.000 in MDD), while receiving novel and maximal input from reward networks (RN→SN: 0.000 in HC vs. 1.000 in MDD). This suggests dysregulated salience attribution with abnormal prioritization of reward-related information—consistent with altered incentive processing in depression.

3. **Reconfigured Limbic Network Inputs:** LIN receives novel regulatory input from frontoparietal networks (FPN→LIN: 0.000 in HC vs. 0.502 in MDD), slightly reduced input from DMN (DMN→LIN: 0.685 in HC vs. 0.614 in MDD), and complete loss of reward-related input (RN→LIN: 0.398 in HC vs. 0.000 in MDD). This pattern suggests compensatory cognitive control over emotional processing with concurrent disconnection from reward systems—potentially reflecting increased regulatory effort and emotional-reward decoupling in depression.

4. **Altered Frontoparietal Control Network Inputs:** FPN receives increased reward network input (RN→FPN: 0.330 in HC vs. 0.475 in MDD) with concurrent reduction in limbic system input (LIN→FPN: 0.586 in HC vs. 0.174 in MDD). This suggests a shift from emotional to reward-related influences on cognitive control processes—potentially reflecting altered motivational influence on executive function in depression.

5. **Reward Network Input Reconfiguration:** RN receives reduced emotional input from limbic networks (LIN→RN: 0.301 in HC vs. 0.150 in MDD) and slightly increased input from frontoparietal networks (FPN→RN: 0.000 in HC vs. 0.082 in MDD). This suggests diminished emotional influence on reward processing with increased cognitive modulation—potentially underlying the cognitive override of natural reward responses in depression.

6. **Global Network Reorganization:** Overall, MDD exhibits a systematic shift in information flow, with increased reward network output to other circuits, emergence of frontoparietal-to-limbic connectivity, reduced limbic network output, and diminished frontoparietal influence on default mode processing. This global reorganization reflects fundamental alterations in the hierarchical processing of self-referential, cognitive, emotional, and reward information.

These patterns reveal a comprehensive reorganization of inter-circuit information reception in MDD, characterized by altered regulatory inputs to self-referential processing, compensatory cognitive control over emotional processing, abnormal reward signal integration, and fundamental disconnection between reward and emotional systems. This circuit-level reconfiguration aligns with core MDD symptoms including negative self-focus, emotional dysregulation, anhedonia, and cognitive control deficits, while providing a neurobiologically grounded framework for understanding depression pathophysiology.

### A.8.4 INTEGRATION OF MULTI-LEVEL INTERPRETABILITY

The three complementary analyses above provide a comprehensive, multi-level interpretation of depression neurobiology through the lens of our NH-GCAT model:

- **Local Level (RG-Fusion):** Frequency-specific analyses demonstrate the model's heightened sensitivity to low-frequency neural oscillations associated with depression, thereby facilitating effective pattern recognition and enhancing classification accuracy.

- **Circuit Level (HC-Pooling):** Hierarchical organization analysis reveals circuit-specific alterations in information processing hierarchy, aligning with clinical manifestations of depression.

- **Network Level (VLCA):** Causal interaction analysis uncovers altered patterns of directed information flow among neural circuits, characterizing the global dysregulation observed in MDD.

This multi-level interpretability not only enhances the model's transparency but also provides mechanistic insights into how local neural abnormalities propagate to circuit-level dysfunction and ultimately manifest as network-level dysregulation in depression.

**Clinical Implications.** The interpretability features of NH-GCAT offer several potential clinical applications:

1. **Biomarker Identification:** The frequency-specific neural patterns identified by RG-Fusion could serve as potential biomarkers for depression diagnosis.

2. **Treatment Targeting:** The circuit-specific hierarchical abnormalities revealed by HC-Pooling could guide targeted interventions such as transcranial magnetic stimulation (TMS) or deep brain stimulation (DBS).

3. **Monitoring Disease Progression:** The causal circuit interactions quantified by VLCA could be used to monitor disease progression and treatment response.

These interpretability analyses demonstrate how NH-GCAT bridges the gap between data-driven machine learning and neuroscientific understanding, offering both predictive power and mechanistic insights into depression pathophysiology.

### A.9 DISCUSSION ON CLINICAL RELEVANCE AND FUTURE DIRECTIONS

Beyond classification accuracy, a primary goal of developing mechanism-aware models like NH-GCAT is to bridge the gap between computational findings and clinical practice. This section discusses the clinical relevance of our model's neurobiological findings, particularly those from the Variational Latent Causal Attention (VLCA) module, and outlines a key future direction in personalized psychiatry.

**Alignment with Known Pathophysiology.** Our VLCA module identified abnormally increased directed information flow from the Reward Network (RN) to the Default Mode Network (DMN) as a significant feature distinguishing individuals with MDD from healthy controls. This finding is highly congruent with established neurobiological theories of depression. It provides a plausible mechanistic link between two core symptom domains: anhedonia (a blunted response to reward, associated with RN dysfunction) and pathological rumination (maladaptive, self-referential thought, associated with DMN hyperactivity). The model's discovery suggests a pathway through which dysfunctional reward signals are pathologically integrated into the brain's self-referential processing stream, perpetuating a cycle of negative self-focus and diminished pleasure.

**Alignment with Treatment Mechanisms.** Crucially, the inter-circuit connections highlighted by our model are not merely statistical artifacts; they represent known targets for antidepressant interventions. The DMN, and its connectivity with other large-scale networks, is a well-established locus of modulation for various treatments, including Selective Serotonin Reuptake Inhibitors (SSRIs). For instance, multiple studies have demonstrated that successful antidepressant treatment is associated with the normalization of DMN connectivity patterns (Dunlop et al., 2017). Therefore, the RN→DMN hyperconnectivity identified by NH-GCAT represents a clinically relevant and treatment-sensitive neurobiological signature, validating that our model is learning features with genuine clinical significance.

**Potential for Predicting Therapeutic Response and Personalized Medicine.** The strong alignment between our model's findings and known treatment mechanisms points directly to a critical future application: predicting individual therapeutic response. While traditional group-level analyses can identify general biomarkers, NH-GCAT can quantify the strength of these directed circuit interactions (e.g., the RN→DMN connection) on a subject-specific basis. This capability allows for the formulation of a precise, testable clinical hypothesis: The baseline magnitude of RN→DMN information flow in a patient, as quantified by our VLCA module, may serve as a predictive biomarker for their response to therapies known to target reward and rumination circuits.

For example, patients exhibiting extreme hyperconnectivity might be predicted to respond more favorably to treatments designed to decouple these systems, such as specific classes of antidepressants, ketamine, or targeted psychotherapies like cognitive behavioral therapy.

Validating this hypothesis requires longitudinal datasets containing pre- and post-treatment neuroimaging data, which was beyond the scope of the current study. Nevertheless, the ability of NH-GCAT to generate such specific, interpretable, and individual-level neurocomputational markers underscores its potential as a tool for advancing personalized psychiatry, moving beyond one-size-fits-all diagnostic labels toward biologically informed, individualized treatment strategies.

## A.10 FURTHER DISCUSSION

**Limitations.** While NH-GCAT demonstrates strong performance and interpretability, several limitations remain. First, the model is trained and evaluated solely on the REST-meta-MDD dataset, which predominantly comprises Chinese participants. This may limit its generalizability to populations with different genetic backgrounds or cultural contexts. Second, depression is inherently heterogeneous, yet our current framework does not distinguish between clinical subtypes due to limited phenotypic information. Third, our neurocircuitry-inspired design relies on predefined circuit definitions from the literature, potentially overlooking individual variability in circuit organization.

**Potential Societal Impacts.** Given that our research involves psychiatric disorder diagnosis, it is important to consider its broader societal implications. NH-GCAT has the potential to enhance our understanding of depression neurobiology and improve diagnostic accuracy, particularly in cases where traditional clinical assessment is challenging. By providing objective, brain-based markers of depression, our approach could help reduce stigma associated with psychiatric disorders and validate patients' experiences. However, as with any AI-assisted diagnostic system, NH-GCAT should be viewed as a complementary tool to support clinical decision-making rather than replace comprehensive psychiatric evaluation. The final diagnostic decisions should always integrate neuroimaging findings with clinical expertise and patient-reported symptoms. As we move toward clinical translation, developing appropriate guidelines for responsible implementation will be essential.

## B  LLM Usage Statement

During the preparation of this manuscript, we utilized a large language model (LLM) as a writing assistance tool. The primary role of the LLM was to aid in polishing the text by improving grammar, clarity, style, and conciseness. The LLM was not used for generating core research ideas, proposing methodologies, conducting experiments, analyzing results, or drawing scientific conclusions. All claims, results, and the scientific narrative remain the original work of the authors, who take full responsibility for all content presented in this paper.

## C  Extended Comparative Analysis and External Generalization

To ensure the robustness of our findings and address potential variations in experimental setups across published works, we conducted two additional rigorous evaluations: (1) a controlled reproduction of baseline methods under identical experimental conditions on the REST-meta-MDD dataset, and (2) an external zero-shot generalization test on an independent dataset (the Japanese Strategic Research Program for the Promotion of Brain Science (SRPBS)) to evaluate cross-dataset transferability.

### C.1  SRPBS Dataset and Preprocessing

To evaluate the generalization capability of NH-GCAT, we utilized the Japanese Strategic Research Program for the Promotion of Brain Science (SRPBS) multi-site dataset. The SRPBS-MDD rs-fMRI dataset comprises $N = 336$ subjects (171 MDD, 165 HC) collected from 5 distinct clinical centers in Japan. This dataset introduces significant domain shifts regarding scanner protocols and population demographics compared to the REST-meta-MDD dataset.

The participant demographics are as follows: The mean age was $42.3 \pm 13.1$ years (range 18–80), with the MDD group averaging $40.8 \pm 10.3$ years and the HC group $43.9 \pm 15.3$ years. The sex distribution was balanced, with 167 males (49.7%) and 169 females (50.3%). For the MDD group, the mean Beck Depression Inventory-II (BDI-II) score was $26.8 \pm 10.7$, indicating moderate to severe depressive symptoms. We applied the same preprocessing pipeline and feature extraction as described in Appendix A.1 to ensure feature alignment.

### C.2  Controlled Baseline Reproduction on REST-meta-MDD

While Table 1 in the main text reports metrics directly from original publications to provide a broad context, Table 11 presents a strictly controlled comparison. Here, we re-evaluated all general graph baselines and a subset of state-of-the-art methods (BPI-GNN, BrainIB, CI-GNN, LCCAF) for which official code was open-source and reproducible in our environment. All models in this comparison were trained using the exact same 5-fold cross-validation splits and hardware setup as NH-GCAT to eliminate variations arising from data partitioning or computational resources.

1. **General Graph Baselines:** GCN, GIN, GraphSAGE, GPS, and GAT, which were implemented within our framework.

2. **Reproducible SOTA Methods:** A subset of specialized MDD classification models (BPI-GNN, BrainIB, CI-GNN, and LCCAF) selected based on the availability and reproducibility of their open-source code.

**Results Analysis.** As shown in Table 11, under these strictly controlled conditions, NH-GCAT continues to demonstrate state-of-the-art performance, achieving the highest AUC (78.5%), Accuracy (73.8%), and F1-score (75.0%). Notably, while simple architectures like GAT achieve high sensitivity (77.5%), they suffer from significant drops in specificity (57.2%), indicating a bias toward positive class prediction. In contrast, NH-GCAT maintains a balanced profile (Sensitivity: 76.4%, Specificity: 71.0%), confirming that our hierarchical causal modeling effectively distinguishes true depressive patterns from healthy controls without overfitting to the majority class or noise. Among the specialized SOTA methods, BrainIB remains the strongest competitor but still lags behind NH-GCAT by 5.9% in AUC and 3.5% in F1-score.

To further visualize the stability of NH-GCAT, Figure 7 displays the Receiver Operating Characteristic (ROC) and Precision-Recall (PR) curves across the 5-fold cross-validation on REST-meta-MDD. The shaded regions represent the standard deviation across folds. The ROC curves (Figure 7a) demonstrate a consistent convex shape with minimal variance, confirming that the model's discriminative power is robust to data partitioning. Similarly, the PR curves (Figure 7b) maintain high precision even at higher recall levels, indicating that the model effectively minimizes false positives—a crucial capability often compromised in imbalanced psychiatric datasets.

Table 11: Performance comparison on the **REST-meta-MDD** dataset comparing NH-GCAT against general graph baselines and **selected reproducible SOTA methods** (5-fold cross-validation). All models were trained and tested on identical data splits. Best results are bolded; second best are underlined.

| Model | AUC | ACC | SEN | SPE | F1 |
|---|---|---|---|---|---|
| *External SOTA models (Re-implemented)* | | | | | |
| BPI-GNN | 70.1 (4.8) | 67.2 (2.9) | 73.4 (8.1) | 60.6 (6.5) | 69.7 (3.7) |
| BrainIB | 72.6 (4.0) | 70.4 (3.4) | 72.0 (6.0) | 68.6 (4.7) | 71.5 (3.8) |
| CI-GNN | 69.5 (4.3) | 66.5 (3.4) | 64.5 (9.6) | 68.6 (11.6) | 66.3 (4.6) |
| LCCAF | 61.8 (3.1) | 62.3 (2.0) | 61.3 (8.2) | 63.3 (9.2) | 62.6 (3.6) |
| *General Graph Baselines* | | | | | |
| GCN | 70.6 (2.4) | 65.8 (1.1) | 67.2 (10.0) | 64.2 (10.1) | 66.8 (4.0) |
| GIN | 70.8 (2.0) | 66.3 (1.9) | 65.7 (14.4) | 67.0 (12.7) | 66.3 (5.2) |
| GraphSAGE | 69.8 (2.6) | 65.6 (1.5) | 64.1 (7.4) | 67.3 (8.5) | 65.8 (2.8) |
| GPS | 67.6 (5.0) | 64.3 (3.9) | 63.3 (16.4) | 65.5 (10.9) | 63.9 (8.4) |
| GAT | 71.5 (3.2) | 67.7 (2.7) | **77.5 (9.1)** | 57.2 (9.4) | 71.2 (3.3) |
| **NH-GCAT (Ours)** | **78.5 (1.7)** | **73.8 (1.4)** | 76.4 (5.8) | **71.0 (6.6)** | **75.0 (1.8)** |
| Improvement | +5.9 | +3.4 | -1.1 | +2.4 | +3.5 |

## C.3 EXTERNAL GENERALIZATION TO SRPBS

To assess clinical utility, we performed a zero-shot evaluation where models trained on REST-meta-MDD were directly tested on the SRPBS dataset without any fine-tuning. This represents a challenging scenario due to significant differences in scanner manufacturers and acquisition protocols between the two datasets.

**Results Analysis.** Table 12 summarizes the external validation performance. As expected, all models experienced a performance drop compared to internal cross-validation, reflecting the domain shift. However, NH-GCAT demonstrated superior generalization capabilities:

- **Overall Discriminability:** NH-GCAT achieved the highest AUC (69.8%) and Accuracy (65.7%), significantly outperforming the next best method (GPS) by +4.2% in AUC and +2.5% in Accuracy. This indicates that the latent representations learned by NH-GCAT are more robust to site-specific noise.
- **Balanced Predictions:** While models like GCN achieved high sensitivity (81.4%), their specificity collapsed to 38.4%, suggesting the model generalized poorly by over-predicting the pathological class. Similarly, BrainIB skewed towards specificity (60.7%) at the cost of sensitivity. NH-GCAT provided the most stable trade-off (Sensitivity: 71.5%, Specificity: 59.8%).
- **Impact of Neurocircuitry Priors:** The superior generalization of NH-GCAT supports our hypothesis that incorporating neurobiological priors (via HC-Pooling) acts as an effective regularizer. By forcing the model to learn interactions between established neural circuits rather than arbitrary node connections, the model focuses on biological signal that is conserved across populations, rather than dataset-specific artifacts.

We provide a visual comparison of the generalization performance in Figure 8, plotting the ROC and PR curves for NH-GCAT against key baselines on the external SRPBS dataset. Despite the significant domain shift, NH-GCAT (red line) maintains a superior envelope over competing methods. In

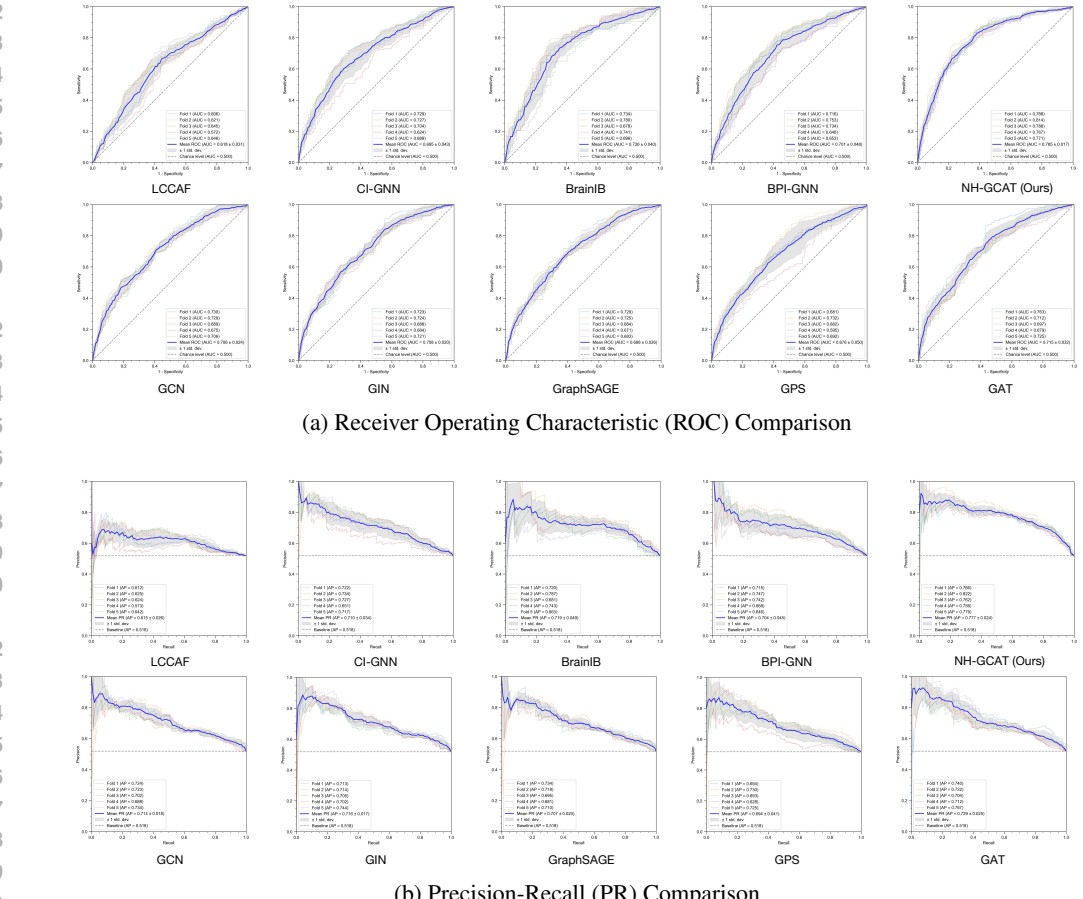

(a) Receiver Operating Characteristic (ROC) Comparison

(b) Precision-Recall (PR) Comparison

Figure 7: **Internal Validation Performance Curves.** Detailed comparison of (a) ROC and (b) Precision-Recall curves for NH-GCAT against baseline models on the REST-meta-MDD dataset (5-fold cross-validation). NH-GCAT demonstrates a dominant area under the curve compared to competing methods, maintaining high precision even at higher recall rates.

the ROC space (Figure 8a), NH-GCAT demonstrates a steeper initial ascent, implying better identification of true positives at low false-positive rates. The PR comparison (Figure 8b) is particularly revealing: while baselines like GCN exhibit a sharp drop in precision as recall increases (indicative of numerous false positives), NH-GCAT sustains a more balanced profile. This visual evidence reinforces that the neurocircuitry-inspired priors help the model learn transferable biological features rather than site-specific noise.

## C.4 Detailed Performance Visualization and Comparison

To address the need for a direct and intuitive comparison of discriminative power, we visualized the overlaid Receiver Operating Characteristic (ROC) and Precision-Recall (PR) curves of NH-GCAT against key baselines (including general GNNs like GAT, GCN, GPS, GraphSAGE, GIN, and specialized models like CI-GNN, BrainIB, BPI-GNN, LCCAF).

### C.4.1 Internal Validation on REST-meta-MDD

Figure 9 presents the performance curves under the strictly controlled 5-fold cross-validation setting on the REST-meta-MDD dataset.

**ROC Analysis (Figure 9a):** NH-GCAT (solid blue line) demonstrates a dominant performance envelope, achieving the highest Area Under the Curve (AUC = 0.785).

Table 12: External validation on the **SRPBS** dataset. Models were trained on REST-meta-MDD and tested on SRPBS (Zero-Shot). Best results are bolded; second best are underlined.

| Model | AUC | ACC | SEN | SPE | F1 |
|---|---|---|---|---|---|
| *External SOTA models* | | | | | |
| BPI-GNN | 61.2 (1.5) | 61.3 (1.5) | 74.9 (8.9) | 47.3 (9.5) | 66.2 (2.9) |
| BrainIB | 64.4 (1.9) | 62.8 (2.4) | 64.8 (7.0) | **60.7 (3.2)** | 63.8 (3.9) |
| CI-GNN | 59.7 (3.8) | 59.6 (1.7) | 64.6 (9.2) | 54.5 (12.5) | 61.8 (2.6) |
| LCCAF | 53.7 (3.7) | 55.7 (1.9) | 58.2 (22.5) | 53.0 (22.8) | 55.3 (11.8) |
| *General Graph Baselines* | | | | | |
| GCN | 61.0 (3.9) | 60.3 (2.2) | **81.4 (9.8)** | 38.4 (7.3) | 67.4 (3.7) |
| GIN | 62.2 (4.1) | 60.8 (2.2) | 76.1 (7.7) | 44.8 (9.1) | 66.3 (2.5) |
| GraphSAGE | 63.7 (2.8) | 61.4 (1.5) | 74.4 (15.8) | 47.9 (14.8) | 65.6 (5.6) |
| GPS | 65.6 (1.2) | 63.2 (0.9) | 78.2 (9.2) | 47.6 (11.1) | **68.2 (2.3)** |
| GAT | 62.0 (8.0) | 60.6 (5.4) | 64.4 (26.5) | 56.6 (19.0) | 60.3 (13.7) |
| **NH-GCAT (Ours)** | **69.8 (2.2)** | **65.7 (1.9)** | 71.5 (12.7) | 59.8 (13.1) | 67.6 (4.2) |
| Improvement | +4.2 | +2.5 | -9.9 | -0.9 | -0.6 |

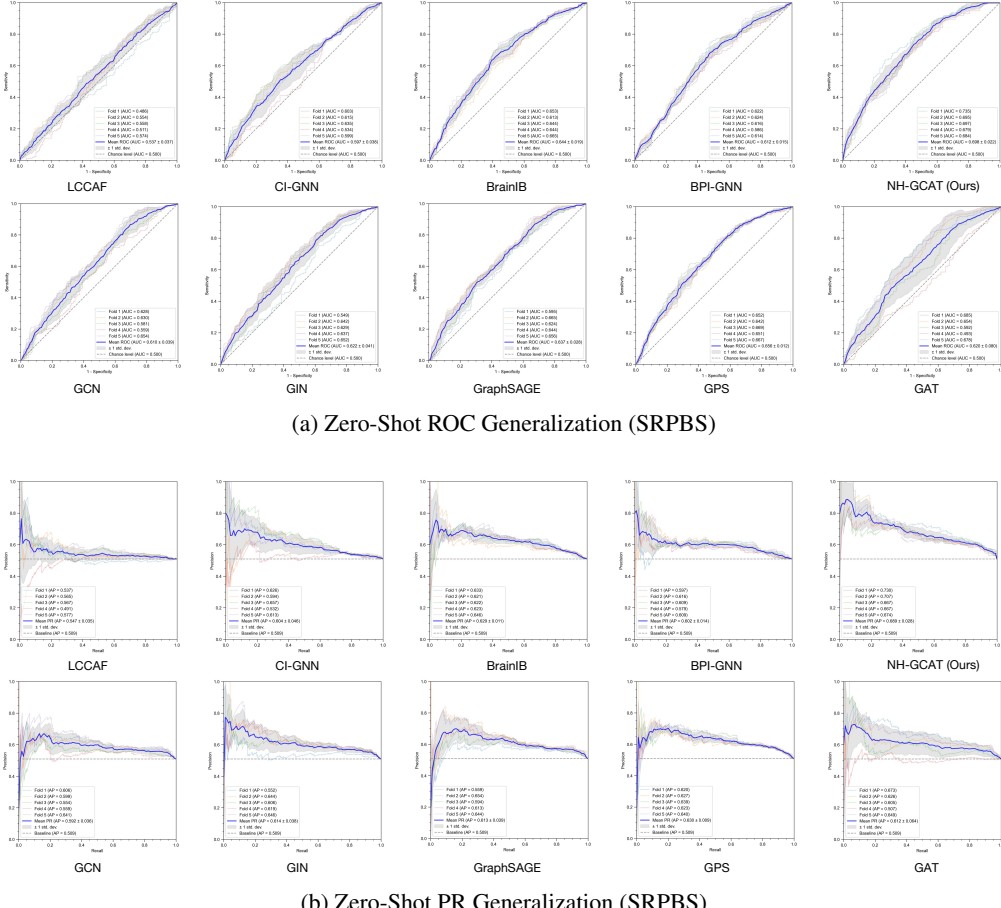

(a) Zero-Shot ROC Generalization (SRPBS)

(b) Zero-Shot PR Generalization (SRPBS)

Figure 8: **External Generalization Performance.** Evaluation of zero-shot transferability on the independent SRPBS dataset. Models trained on REST-meta-MDD were tested directly on SRPBS. (a) ROC curves and (b) Precision-Recall curves show that NH-GCAT maintains a superior performance envelope compared to baseline methods, indicating stronger robustness to site-specific variations and scanner effects.

- **Early Detection Capability:** Crucially, NH-GCAT exhibits a significantly steeper ascent in the high-specificity region (x-axis: $0.0 - 0.2$). At a strict False Positive Rate (FPR) of 0.2, NH-GCAT achieves a sensitivity of approximately 0.65, whereas the strongest baselines (e.g., BrainIB, pink line; BPI-GNN, brown line) struggle to surpass 0.55. This indicates that NH-GCAT is far more effective at identifying positive cases while minimizing misdiagnoses.

- **Baseline Comparison:** While methods like BrainIB (AUC = 0.726) and GAT (AUC = 0.715) show competitive performance, they are consistently enclosed by the NH-GCAT curve. Notably, methods like LCCAF (yellow line) and GPS (light blue line) show limited discriminative power with substantially lower AUCs (0.618 and 0.676, respectively).

**Precision-Recall Analysis (Figure 9b):** The Precision-Recall curves further corroborate the robustness of our model, with NH-GCAT achieving the highest Average Precision (AP = 0.777).

- **Stability of Precision:** As Recall increases, NH-GCAT maintains a superior Precision level compared to all baselines. For instance, at a Recall of 0.8, NH-GCAT sustains a Precision above 0.7, whereas most baselines drop below 0.65.

- **Robustness to False Positives:** The gap between NH-GCAT and the cluster of baselines (e.g., GIN, CI-GNN) highlights that our neurocircuitry-inspired architecture effectively reduces false positive predictions even when the decision threshold is relaxed.

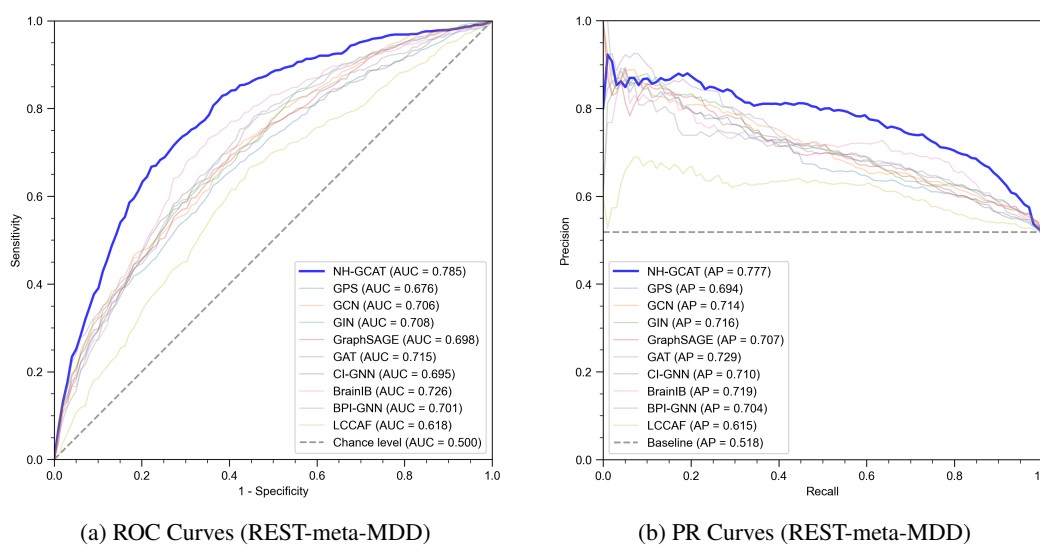

(a) ROC Curves (REST-meta-MDD)      (b) PR Curves (REST-meta-MDD)

Figure 9: **Overlaid Performance Curves on REST-meta-MDD (Internal 5-fold CV).** (a) NH-GCAT achieves the highest AUC (0.785), showing superior sensitivity at low false positive rates. (b) The model maintains the highest Average Precision (AP = 0.777), indicating stable performance across decision thresholds.

### C.4.2 EXTERNAL GENERALIZATION ON SRPBS

To rigorously evaluate clinical transferability, Figure 10 presents the overlaid curves on the independent **SRPBS** dataset. These results represent a **zero-shot** setting, where models trained on REST-meta-MDD were applied directly to SRPBS without any fine-tuning.

**ROC Analysis (Figure 10a):** Despite significant domain shifts caused by different scanner protocols, NH-GCAT (solid blue line) maintains a distinct performance advantage.

- **Robustness to Domain Shift:** While most baseline methods (e.g., LCCAF, CI-GNN) suffer from severe performance degradation—with curves flattening towards the diagonal chance line, NH-GCAT preserves a convex shape, achieving the highest AUC of 0.698. This indicates that the neurocircuitry-inspired features are biologically invariant rather than site-specific artifacts.

- **Comparison with Strong Baselines:** Even compared to the best-performing baseline (GPS, light blue line, AUC = 0.656), NH-GCAT shows a consistent margin of improvement. Notably, in the critical low-FPR region ($x < 0.2$), **NH-GCAT demonstrates a significantly steeper ascent compared to the cluster of baseline methods, establishing a clear performance margin even against the strongest competitors.**

**Precision-Recall Analysis (Figure 10b):** The PR curves highlight the challenge of the zero-shot task yet confirm NH-GCAT's stability.

- **Superior Precision Envelope:** NH-GCAT achieves the highest Average Precision (AP = 0.689), significantly outperforming the next best method (BrainIB, AP = 0.629). As shown in the plot, the NH-GCAT curve consistently stays above all others.
- **Reliability at High Sensitivity:** A key observation is the "tail" of the PR curve. At high recall levels ($> 0.8$), where most models converge to the baseline prevalence (grey dashed line), NH-GCAT maintains higher precision. This suggests that even when pushed to identify the majority of patients in a new dataset, our model introduces fewer false positives than competing methods.

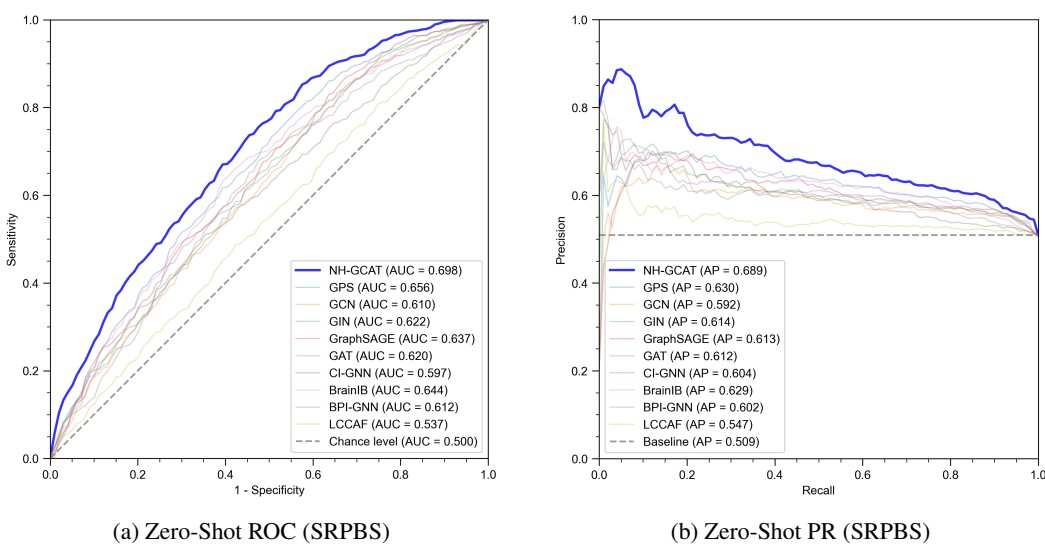

(a) Zero-Shot ROC (SRPBS)  (b) Zero-Shot PR (SRPBS)

Figure 10: **Overlaid Generalization Curves on SRPBS (Zero-Shot Transfer).** Models trained on REST-meta-MDD were tested directly on SRPBS without fine-tuning. (a) NH-GCAT (Red) significantly outperforms baselines, demonstrating superior robustness to domain shifts. (b) Precision-Recall curves confirm that NH-GCAT offers the most reliable clinical utility in unseen domains.

