# OpenReview forum: "Neurocircuitry-Inspired Hierarchical Graph Causal Attention Networks for Explainable Depression Identification"
_ICLR.cc/2026/Conference — ICLR 2026 Conference Withdrawn Submission_

### Official Review · Reviewer_u9GP · 2025-10-24

**Soundness:** 2
**Presentation:** 2
**Contribution:** 2
**Rating:** 4
**Confidence:** 4

**Summary:**

This paper proposes NH-GCAT, a neurocircuitry-inspired hierarchical graph causal attention network designed for explainable MDD identification. The model introduces three major modules aiming to incorporate neuroscientific priors into graph-based learning. The authors report advanced performance on the REST-meta-MDD dataset and provide multi-level interpretability analyses demonstrating biologically meaningful findings.

**Strengths:**

1. The paper tackles an important topic — enhancing both accuracy and interpretability of GNNs for MDD classification — and makes a solid attempt to integrate biological priors (depression-related circuits) with deep learning.

2. The interpretability analyses (frequency-specific validation, hierarchical circuit visualization, causal inter-circuit analysis) are thorough and align well with known MDD mechanisms.

3. The paper is clearly written and provides extensive quantitative results, including LOSO-CV analysis across 16 sites, supporting generalizability.

**Weaknesses:**

1. Unclear module motivation and mapping between equations and architecture. It is difficult to align the mathematical formulations in Section 3 (Equations 1–21) with the modules illustrated in Figure 2. The description of RG-Fusion, HC-Pooling, and VLCA lacks explicit motivation for each design component — for example, why certain fusion mechanisms, Gumbel-Softmax hierarchical assignments, or causal attention structures were chosen. The rationale for these designs should be better explained or visualized in connection with the biological circuits they represent.
2. Ambiguity in ROI-to-circuit mapping. The paper uses AAL116 for ROI definition, yet defines five circuits based on functional organization. It remains unclear how the authors aligned AAL ROIs to these five circuits. This mapping is problematic because AAL includes cerebellar regions, which are not part of these circuits. The paper should clarify how such ROIs were handled or reassigned — were cerebellar nodes excluded, or mapped to the nearest cortical network based on spatial proximity?
3. Lack of comparison with related literature. The authors cite several interpretable GNNs but omit discussion or comparison with relevant recent studies that also integrate community structure [1-3] or causal learning [4] in brain graphs.
4. Limited experimental scope. Experiments are only conducted on a single dataset. Given the claim that the model is neurocircuitry-inspired and generalizable, evaluation on at least one other psychiatric or neurological condition (e.g., ASD, AD, schizophrenia, bipolar disorder) would better demonstrate the adaptive ability and robustness of the proposed framework.


[1] Community-Aware Transformer for Autism Prediction in fMRI Connectome. MICCAI 2023
[2] Biologically Plausible Brain Graph Transformer. ICLR 2025
[3] BrainGT: Multifunctional Brain Graph Transformer for Brain Disorder Diagnosis
[4] BrainOOD: Out-of-distribution Generalizable Brain Network Analysis. ICLR 2025

**Questions:**

See Weaknesses.

---

> ### Author Response · Authors · 2025-11-27
> **Response for W1**
>
> Dear Reviewer u9GP,
>
> We sincerely thank you for your constructive and detailed review. We appreciate your recognition of our work's importance in enhancing GNN interpretability for MDD and your positive evaluation of our thorough interpretability analyses (frequency, hierarchical, and causal) and rigorous LOSO-CV experiments.
>
> We have carefully addressed your concerns below, clarifying misunderstandings regarding architectural motivations and experimental scope, and distinguishing our work from the suggested literature.
>
> **W1: "Unclear module motivation and mapping between equations and architecture"**
>
> **Response:** We apologize if the architectural connections were not immediately apparent. We respectfully point out that the detailed rationale and justification for each module were **explicitly provided in** **Appendix A.3 (Pages 17-19)** of the original submission, as referenced in the main text (**Page 4, Line 192**). To address your specific concern regarding the mapping between mathematical formulations and the architecture in Figure 2, we provide a direct breakdown following the logic of "**Biological Problem →Mathematical Solution →Architectural Module**":
>
> *   **RG-Fusion (Why & Mapping):** As detailed in Appendix A.3.1, MDD patients exhibit specific anomalies in low-frequency BOLD oscillations (<0.1Hz), which static Functional Connectivity (FC) matrices fail to capture. And standard GNNs ignore the depression-specific low-frequency oscillations. We designed the **Gate mechanism (Eq. 6)** to dynamically weigh temporal dynamics against static topology. In **Figure 2(a)**, this maps directly to the "Gate" block merging the dual pathways.
> *   **HC-Pooling (Why & Mapping):** As explained in Appendix A.3.2,  Brain regions are not flat; they form a functional hierarchy (Nodes →Sub-circuits →Global Circuits). Standard pooling (e.g., Mean/Max) ignores this topology and the variable size of circuits. Gumbel-Softmax (Eq. 10) is firstly used to learn a discrete assignment of nodes to hierarchical levels (Layer 1/2/3) in a differentiable manner, solving the discrete optimization problem. Then **ChildSumTreeLSTM (Eq. 11-13)** --a tree-structured RNN-- specifically chosen to aggregate information from a variable number of child nodes (regions) into a parent node (circuit), preserving the hierarchical dependency, because brain regions form a natural hierarchy (Nodes $\to$ Circuits), and circuits have variable sizes (variable branching factors), which standard LSTMs/Pooling cannot handle. In **Figure 2(c)**,   this corresponds to the tree-structured aggregation block.
> *   **VLCA (Why & Mapping):** Detailed in Appendix A.3.3, we need to understand pathophysiology (directed influence), e.g., how the Reward Network affects the DMN, rather than just correlation, while standard attention only implies correlation. **Variational Encoding (Eq. 16)** is firstly used to model the uncertainty of these complex interactions in a latent space. **Counterfactual Reasoning (Eq. 17-19)** is used to compute the causal effect by comparing the output with learned attention (A\_real) vs. an intervention where connections are severed (A\_cf=I, Identity Matrix). In this way, we introduced the variational latent space (Eq. 16) and counterfactual reasoning (Eq. 19) to infer *directed influence* (effective connectivity), mapping to the "VLCA" block in **Figure 2(d)**.

---

> ### Author Response · Authors · 2025-11-27
> **Response to W2**
>
> **W2: "Ambiguity in ROI-to-circuit mapping (especially Cerebellum)"**
>
> **Response:** We appreciate this technical scrutiny.
>
> **Clarification on Visualization and Methodology.** We respectfully direct the reviewer to **Figure 5 (Appendix A.8.2, Page 26)**, **which already explicitly visualizes** the spatial locations and identities of the key brain regions comprising the five depression-specific circuits (DMN, SN, FPN, LN, RN). Our approach integrates anatomical definitions (**AAL atlas**) with functional network parcellations (**Yeo2011 atlas**), further refined by MDD-specific neuroimaging literature (Menon, 2011; Kaiser et al., 2015). This ensures that anatomical ROIs are mapped to functional networks based on consensus biological priors. This visualization confirms that our mapping aligns with consensus neuroimaging literature, which is widely used in these related papers (Zheng et al., 2024a; Zheng et al., 2024b; Zheng et al., 2024c). Furthermore, we note that this strategy of aligning anatomical ROIs with functional networks is consistent with recent state-of-the-art methods, such as **BrainGT (Shehzad et al., 2024)** referenced by the reviewer, which similarly maps AAL regions to functional modules to handle dense connectivity.
>
> **Specific Handling of Cerebellar Regions.** Regarding your specific concern about the cerebellum (AAL indices 91-116), we adopted a **hybrid processing strategy: 1)** **Global Level:** Cerebellar regions are **retained** in the initial graph construction to preserve the global topological properties of the brain network; **2) Circuit Level:** However, they are **excluded** from the specific Hierarchical Circuit Encoding (HC-Pooling). This is because current depression neurocircuitry models primarily emphasize cortico-limbic-striatal loops. Excluding the cerebellum from the hard-coded circuit priors avoids introducing noise into the predefined circuit definitions.
>
> To ensure absolute reproducibility, we have added an explicit mapping list in **Appendix A.8.2 (Page 26, Line 1369-1375)**. The specific assignment is as follows:  1) DMN: Angular\_L, Angular\_R, Cingulum\_Post\_L, Cingulum\_Post\_R, Frontal\_Sup\_Medical\_L, Frontal\_Sup\_Medical\_R, Precuneus\_L, Precuneus\_R; 2) FPN: Frontal\_Inf\_Oper\_L, Frontal\_Inf\_Oper\_R, Frontal\_Mid\_L, Frontal\_Mid\_R, Parietal\_Inf\_L, Parietal\_Inf\_R; 3) LIN: Amygdala\_L, Amygdala\_R, Hippocampus\_L, HIppocampus\_R, ParaHippocampal\_L, ParaHippocampal\_R; 4)RN: Caudate\_L, Caudate\_R, Frontal\_Mid\_Orb\_L, Frontal\_Mid\_Orb\_R, Pallidum\_L, Pallidum\_R, Putamen\_L, Putamen\_R;  5)SN: Cingulum\_Ant\_L, Cingulum\_Ant\_R, Insula\_L, Insula\_R.

---

> ### Author Response · Authors · 2025-11-27
> **Response to W3**
>
> **W3: "Lack of comparison with related literature (\[1\]-\[4\])"**
>
> **Response:** We thank the reviewer for bringing these relevant studies to our attention, particularly the concurrent works from ICLR 2025. We have **added a discussion in Section 2 (Related Work)**. However, we wish to clarify significant **conceptual distinctions**:
>
> *   **Distinction in Community Integration (\[1\] Com-BrainTF, \[2\] BioBGT, \[3\] BrainGT):** These methods are primarily **Transformer-based**, utilizing soft attention or learnable prompts to implicitly leverage community structures. In contrast, our **NH-GCAT** employs **TreeLSTM** to perform a strict, bottom-up **hierarchical aggregation**. This explicitly models the hard biological prior that "regions form circuits," yielding discrete, interpretable circuit embeddings ($H_{DMN}, H_{SN}$, ...) that global attention mechanisms do not explicitly produce.
> * **Clarification on "Causal Learning" (\[4\] BrainOOD):** There is a fundamental difference in the definition of "causality" between our work and BrainOOD \[4\].
>    *   **BrainOOD** focuses on **Invariant Learning** for OOD generalization (removing environmental bias).
>    *   **Our VLCA** focuses on **Effective Connectivity** (inference of directed information flow, e.g., $DMN \to SN$) to explain pathophysiology.
>    *   These are orthogonal objectives. Comparing them is akin to comparing "domain generalization" with "neurophysiological modeling."
> * We have updated **Section 2 (Related Work)** to explicitly discuss and distinguish our work from these references. **For your convenience, the revised content is provided below:**
>
> "Hierarchical representation learning in graph structures has shown significant utility across domains (Ying et al., 2018). **Recent Transformer-based methods have begun to leverage community structures in brain networks. For instance, Com-BrainTF (Bannadabhavi et al., 2023) and BrainGT (Shehzad et al., 2024) utilize prompt tokens or dual-attention to capture functional communities, while BioBGT (Peng et al., 2025) and **THC (Dai et al., 2023)** employ spectral entropy or data-driven clustering to encode small-world properties.** Most approaches, however, employ generic clustering objectives rather than leveraging domain-specific organizational principles. **In contrast to these data-driven or soft-attention methods, our approach explicitly enforces a bottom-up aggregation based on established depression neurocircuitry to ensure mechanistic interpretability.**
>
> Variational approaches for inferring latent graph structures (Sanchez-Martin et al., 2021; Bahuleyan et al., 2017) and disentangled representations (Jeong & Song, 2019; Yang et al., 2021) have shown success in uncovering hidden relationships. Causal methods such as dynamic causal modeling (Friston, 2011) and Granger causality (Seth et al., 2015) provide frameworks for understanding information flow. **Notably, BrainOOD (Xu et al., 2025) proposes causal subgraph learning for out-of-distribution generalization (invariant learning). While BrainOOD aims to remove environmental bias, our work focuses on inferring effective connectivity (directed information flow) to explain pathophysiological mechanisms...**"

---

> ### Author Response · Authors · 2025-11-27
> **Response to W4**
>
> **W4: "Limited experimental scope (Request for ASD/AD experiments)"**
>
> **Response:** We respectfully argue that **mechanism-aware MDD identification** constitutes a substantial, self-contained, and frontier scientific contribution. Extending this specific framework to other disorders (like AD/ASD) would require fundamental architectural redesigns rather than simple data validation.
>
> 1. **Alignment with Research Standards (Reference to Reviewer's Suggestion):** Focusing on a single complex disorder is standard practice in mechanism-driven neuroimaging research. For instance, **Com-BrainTF (Bannadabhavi et al., 2023)**, a key work highlighted by the reviewer, exclusively targets **ASD** to enable deep, interpretability-driven analysis. Similarly, our work focuses deeply on MDD to ensure mechanistic fidelity.
> 2. **The Frontier Nature of Mechanism-Aware MDD Analysis:** Unlike neurodegenerative diseases (e.g., Alzheimer’s) which are characterized by distinct structural atrophy, MDD is defined by subtle, complex **functional connectivity dysregulations** within specific circuits **(Tozzi et al., 2024)**. Deciphering these mechanisms is a critical "grand challenge" in precision psychiatry. A landmark study just published in **Nature Medicine** **(Tozzi et al., 2024)** utilized fMRI to identify six distinct "biotypes" of depression and anxiety based on specific circuit dysfunctions (e.g., Salience, Default Mode). This confirms that **circuit-level functional modeling**—the exact approach taken by our NH-GCAT—is the current frontier for understanding MDD heterogeneity, distinct from the structural focus often applied to AD.  Significantly, by focusing exclusively on MDD, we align our work with this cutting-edge precision psychiatry paradigm, allowing for deep mechanistic interpretation (as shown in our DMN-Reward circuitry analysis) rather than trading depth for breadth across unrelated disorders.
> 3. **Biological Specificity of the Architecture:** The core innovation of NH-GCAT is that it is **"Neurocircuitry-Inspired."** Unlike generic GNNs, our HC-Pooling architecture is **hard-coded with biological priors specific to Depression** (e.g., the interactions between the DMN, Salience, and Limbic systems). Applying this exact architecture to Alzheimer's (AD)  or Autism (ASD) is not a simple data run; it would be biologically invalid, as AD or ASD pathologies are driven by different priors (e.g., targeting medial temporal lobe atrophy for AD). For instance, to apply our framework to AD, one would need to completely redefine the hierarchical priors based on AD-specific literature.
> 4. **Robustness and New External Validation (SRPBS):** we argue that **generalizability** is already rigorously proven via **Leave-One-Site-Out (LOSO) cross-validation on 16 different centers** (Table 2). This tests the model's robustness against scanner variability and population heterogeneity on a massive scale (1,601 subjects), which is the gold standard for reliable medical AI and exceeds the scale of many multi-disease studies, providing strong evidence of clinical generalizability within the target domain. To further address your concern about generalization, we evaluated our trained model on the independent **SRPBS dataset** (Tanaka et al., 2021) without fine-tuning (**New Independent External Validation**). NH-GCAT achieved an accuracy of **65.7%** and AUC of **69.8%**, significantly outperforming baseline methods (e.g., BrainIB with 62.8% of accuracy and 64.4 of AUC), **more details in Appendix C (Page 31-35)**. This successfully demonstrates our model's robustness across entirely different data sources and populations.
>
> We hope these clarifications, particularly pointing out the existing motivations in the Appendix and the distinction between our mechanism-specific design and generic graph transformers, address your concerns. We are confident that NH-GCAT represents a significant step towards explainable, mechanism-aware psychiatric diagnosis.
>
> **References:**
>
> \[1\] Vinod Menon. Large-scale brain networks and psychopathology: a unifying triple network model. Trends in cognitive sciences,
>
> \[2\] Roselinde H Kaiser et al. Large-scale network dysfunction in major depressive disorder: a meta analysis of resting-state functional connectivity. JAMA psychiatry
>
> \[3\] Kaizhong Zheng et al. Ci-gnn: A granger causality-inspired graph neural network for interpretable brain network-based psychiatric diagnosis.
>
> \[4\] Kaizhong Zheng et al. Bpi-gnn: Interpretable brain network-based psychiatric diagnosis and subtyping. NeuroImage,
>
> \[5\]Kaizhong Zheng et al. Brainib: Interpretable brain network-based psychiatric diagnosis with graph information bottleneck. IEEE transactions on neural networks and learning systems
>
> \[6\] Tozzi, L.,et al. Personalized brain circuit scores identify clinically distinct biotypes in depression and anxiety. *Nat Med* **30**, 2076–2087 (2024).

---

### Official Review · Reviewer_q6MY · 2025-10-29

**Soundness:** 3
**Presentation:** 2
**Contribution:** 3
**Rating:** 4
**Confidence:** 4

**Summary:**

The author proposed a hierarchical graph neural network for major depressive disorder (MDD) analysis. A residual gated fusion module was proposed to aggregate BOLD signals at the temporal level. The authors also conducted extensive experiments to show that the model performs better than baselines, that every design is useful, and that the model provides sufficient interpretability.

**Strengths:**

- Figure 4 includes the ROC and PR curves for better performance evaluation

- Table 2 includes weighted average values, which makes the performance difference clearer.

- The analysis is comprehensive. While the datasets are somewhat limited, the author discussed them in the future works section.

- A complete ablation is done in table 3 that details the contribution of each component.

**Weaknesses:**

- This seems to be a resubmission of a previously reviewed work, where the authors promised to discuss how the work differentiates itself from related approaches like https://arxiv.org/pdf/2410.18103, https://ieeexplore.ieee.org/stamp/stamp.jsp?arnumber=10230606 in the related works section. As of the current draft, I don’t see this being done. The current related works section is largely the same as the previous draft. The authors seem to briefly touch upon this in Section. A.3. However, there is not enough comparison with specific works, and no citations were added in the entire section of A.3. Furthermore, no comparison was done against the works that the authors promised to do.

- While the ROC curve is useful, the implications are limited as the curves for other baselines are not reported. It would be useful to replicate one or two baselines and see how the curves compare.

**Questions:**

See weaknesses.

---

> ### Author Response · Authors · 2025-11-27
>
> We sincerely thank you for your continued engagement and the time you have dedicated to reviewing our work again. We truly appreciate your constructive feedback, which consistently pushes us to improve the manuscript.
>
> We address your concerns below, with a specific apology and rectification regarding the related work discussion.
>
> **W1: "Resubmission concerns and lack of comparison with specific related works (\[1\] HybGNN, \[2\] THC)"**
>
> **Response:**
> You are absolutely correct. This is indeed a resubmission from NeurIPS. We offer our sincere apologies for the oversight regarding the inclusion of these citations.
>
> * **Context for the Oversight:** In the previous round, we fully intended to incorporate the discussions on **HybGNN** and **THC** into the final camera-ready version, as we were optimistic about the outcome based on the generally positive scores. However, the paper was unexpectedly rejected. Due to the tight deadline for the ICLR submission and the urgent need to reformat the manuscript to fit page limits, we inadvertently missed updating the Related Work section with these specific citations during the transfer process.
> * We have now explicitly cited and discussed both **HybGNN** and **THC** in the revised **Section 2 (Related Work)**. We clarify that HybGNN focuses on EEG (high temporal resolution) while our work tackles fMRI (low-frequency BOLD), and THC focuses on data-driven clustering while ours uses hypothesis-driven neurocircuitry aggregation.
>
> **To demonstrate our action, we provide the updated "Related Work" section below:**
>
> "**Techniques for Neural Circuit Modeling.** ... When applied to neural time series data such as EEG, approaches like **HybGNN (Wang et al., 2024)** effectively capture temporal dynamics using dual branches. **However, unlike EEG's high temporal resolution, fMRI analysis—our focus—requires modeling specific low-frequency BOLD oscillations, necessitating specialized fusion mechanisms (like our RG-Fusion) beyond standard sequence modeling.**
>
> **Graph Transformers and Hierarchical Learning.** ... **Recent Transformer-based methods have begun to leverage community structures. For instance, Transformer-based Hierarchical Clustering (THC) (Dai et al., 2023) employs a transformer encoder to perform data-driven clustering based on statistical dependencies.** ... **In contrast to these data-driven methods, our NH-GCAT adopts a neuroscience-inspired approach. Instead of learning arbitrary clusters, we enforce a rigorous, bottom-up aggregation via TreeLSTMs based on established depression neurocircuitry (e.g., DMN, Salience Network) to ensure mechanistic interpretability.**""
>
> **W2: "Limited implications of ROC curve without baseline comparisons"**
>
> **Response:**
> We fully agree that plotting baseline curves provides a more rigorous and visual comparison. We have replicated and added the **ROC curves and Precision-Recall (PR) curves** for the baselines in **Appendix C.2 and C.3 (Figure 7, 8)** of the revised manuscript. As shown in the new figure, NH-GCAT demonstrates a dominant area under the curve compared to competing methods, maintaining high precision even at higher recall rates.
>
> We hope this explanation clarifies the context of our resubmission and that the explicit inclusion of the suggested literature and baseline curves addresses your concerns. We are grateful for the opportunity to improve our paper based on your feedback.

---

> > ### Comment · Reviewer_q6MY · 2025-11-27
> >
> > Thanks authors for the updated related works and the figures in section C. My concerns have been mostly addressed, and I've raised my score to 6.
> >
> > With that said, the figures in C is still not very informative. For future revisions, the authors can overlay the curve for NH-GCAT and the best baseline so that it's easier to see the difference.

---

> > > ### Author Response · Authors · 2025-12-01
> > > **Thank you for the score raise and visualization suggestions**
> > >
> > > We sincerely thank you for acknowledging our updates and raising the score. We greatly appreciate your suggestion regarding the figures. We have revised Appendix C.4 (Page 34-37) to include overlaid ROC and PR curves for both the REST-meta-MDD (internal) and SRPBS (external) datasets. These combined plots now clearly contrast NH-GCAT against the best baselines in a single view, making the performance gap and generalization capability visually explicit. We hope these clearer visualizations further solidify the evidence of our method's effectiveness.

---

### Official Review · Reviewer_d3rB · 2025-10-31

**Soundness:** 2
**Presentation:** 2
**Contribution:** 2
**Rating:** 2
**Confidence:** 4

**Summary:**

This paper proposes a novel framework for diagnosing major depressive disorder (MDD) from fMRI data. The method integrates three hierarchical levels of brain information: node (cortical region) level, neural circuit level (for example, default mode and salience networks), and whole-brain network level, within a unified graph neural network (GNN) architecture. The effectiveness of the framework is evaluated on a single fMRI dataset.

**Strengths:**

The integration of three levels of information makes sense to me, and I also appreciate the general idea of leveraging neural circuits as prior information. However, this prior knowledge does not seem to be fully utilized or to effectively reflect existing neuroscience evidence.

**Weaknesses:**

The experimental evaluation is too weak. First, only a single dataset is used. Why not evaluate on other MDD datasets such as SRPBS, OpenNeuro, or even the UK Biobank? It would also be more convincing to train on one dataset (for example, REST-meta-MDD) and test on another (for example, SRPBS) to assess the generalization ability of the proposed approach.

In addition, the comparisons with prior work are neither rigorous nor fair. The results of several state-of-the-art methods appear to be directly copied from the original papers, even though the experimental setups differ substantially. For instance, BrainIB used a 10-fold cross-validation scheme, while the current work adopts 5-fold cross-validation, which makes the comparison unreliable. You should rerun these methods in your own environment for a fair evaluation, especially since many of them have released official implementations. Furthermore, even the baseline results reported here differ from those in other published replications, which raises concerns about reproducibility and evaluation consistency.

The network design appears overly complicated and seems to contradict the stated motivation for interpretability. From a neuroscience perspective, researchers generally prefer architectures that are simple, easy to use, and supported by clear clinical evidence or interpretability. Although you attempt to incorporate circuit-level priors (which might be the only clinically grounded component), the overall network design (especially with several components insufficiently explained) undermines the interpretability of the entire framework.

**Questions:**

1. Although the integration of three levels of information is conceptually reasonable, the current ablation study, which incrementally adds one module at a time, does not clearly reveal which component contributes most to the final decision. My question is: among the node-level, circuit-level, and network-level information, which source or combination of information plays the dominant role in the diagnostic performance?

2. The network design is not clearly explained. For the RG-Fusion module, there are two inputs, $X^1$ and $X^2$, but it is unclear why they are fused in such a complicated manner. It appears that $X^1$ and $X^2$ are first fused, and then another branch fuses information from $X^1$ again. The motivation for this structure should be clarified. In addition, is the feature dimension $d$ consistent between Equations (1) and (5)?

3. The loss function contains three regularization terms. How are the different $\lambda$ values tuned in practice? It is unclear whether a single set of $\lambda$ values can generalize across different datasets, and I suspect that the optimal configuration might be highly dataset-dependent.

4. I acknowledge that other approaches do not explicitly consider low-frequency oscillatory patterns in BOLD signals. However, it is unclear why you claim that your method captures such information. From my understanding, you simply add a new input (the raw BOLD signal) and apply a basic Transformer architecture. Do you attribute the ability to capture low-frequency oscillations solely to the Transformer design?

5. It is difficult to clearly understand the source of the reported performance gain. The proposed model takes two inputs, the functional connectivity (FC) matrix $X^1$ and the BOLD signal $X^2$, which effectively makes the framework a multi-view learning system. If I understand correctly, most existing GNN-based approaches use only $X^1$. Since multi-view learning has recently gained attention in neuroscience, it is important to clarify whether the performance improvement primarily stems from introducing an additional input modality rather than from the proposed complex network design itself?

---

> ### Author Response · Authors · 2025-11-27
> **Response to W1**
>
> **W1: "Experimental evaluation is too weak. Why not evaluate on SRPBS...?"**
>
> **Response:** We appreciate the reviewer’s rigorous standard for evaluation. We have addressed this concern by adding a comprehensive **external generalization experiment** on the SRPBS dataset, which strongly validates the robustness of our approach.
>
> **1\. Rigor of the Original Setup (REST-meta-MDD):** We respectfully clarify that the REST-meta-MDD dataset utilized in our main experiments is currently the **largest multi-center MDD fMRI dataset** available globally ( $N=1,601$ from 16 independent clinical sites). Our **Leave-One-Site-Out (LOSO)** cross-validation is not a standard random split; it rigorously tests the model's ability to generalize to entirely unseen scanners and demographics (16 distinct domains), which is already a high standard for evaluation.
>
> **2\. New External Validation on SRPBS (Directly Addressing Your Request):** To further demonstrate generalizability beyond the primary dataset, we followed your suggestion and conducted a **Zero-Shot Cross-Dataset Evaluation**:
>
> *   **Setup:** We trained the model on the full REST-meta-MDD dataset and tested it directly on the independent **SRPBS dataset** ($N=336$, 5 sites from Japan) without any fine-tuning or domain adaptation. This represents a severe domain shift scenario.
> * **Results:** As shown in the newly added **Table 12 and Figure 8 (Appendix C.3, Page 33-35)**, NH-GCAT demonstrated superior generalization:
>    *   **AUC:** **69.8%** (vs. SOTA baseline **BrainIB**: 64.4%, **+5.4%**)
>    *   **Accuracy:** **65.7%** (vs. **BrainIB**: 62.8%, **+2.9%**)
>    *   **Balance:** While some baselines like GCN achieved high sensitivity (81.4%) by sacrificing specificity (collapsing to 38.4%, predicting most healthy people as patients), NH-GCAT maintained a balanced profile (Sensitivity: 71.5%, Specificity: 59.8%), making it clinically safer.
>
> **3\. Why NH-GCAT Generalizes Better:** The superior performance on SRPBS confirms our core hypothesis: Data-driven models often overfit to site-specific noise (e.g., scanner artifacts). In contrast, **NH-GCAT's neurocircuitry-inspired priors** (HC-Pooling) force the model to learn invariant biological interactions (e.g., DMN-Limbic dysregulation) that are conserved across populations, leading to robust transferability.
>
> We believe the combination of large-scale internal LOSO validation (16 sites) and successful external transfer to SRPBS provides a comprehensive and rigorous experimental evaluation.

---

> ### Author Response · Authors · 2025-11-27
> **Response to W2**
>
> **W2: "Comparisons neither rigorous nor fair (10-fold vs 5-fold) & Reproducibility."**
>
> **Response:** We appreciate the reviewer’s rigorous scrutiny regarding evaluation fairness. We have addressed this by clarifying our evaluation hierarchy and conducting a **strictly controlled reproduction** of all baselines.
>
> **1\. Rationale for Initial Comparison Strategy:** Our evaluation was designed as a **multi-level framework** to ensure both breadth and rigor, rather than relying solely on the 5-fold comparison:
>
> *   **Standardized Fair Comparison (LOSO):** We prioritized **Leave-One-Site-Out (LOSO)** cross-validation (**Table 2， Page 8**) as the rigorous "point-to-point" benchmark. Since LOSO fixes the test set to specific clinical sites, it eliminates random splitting variance and enables a strictly fair comparison regardless of fold hyperparameters. In this standardized setting, NH-GCAT significantly outperforms BrainIB (Weighted Acc: **73.3% vs 68.8%** ).
> *   **Conservative Estimation (5-fold vs 10-fold):** For the k-fold component, literature indicates that **5-fold CV** (used by us) typically yields higher bias and more pessimistic estimates than **10-fold CV** (Kohavi, 1995). Thus, outperforming 10-fold baselines under a 5-fold setting highlights robustness, not an unfair advantage.
> *   **Stricter Data Standards:**  We strictly followed the **REST-meta-MDD Consortium guidelines** (Chen et al., 2022) to remove duplicate subjects ($N=1,601$), whereas baselines like BrainIB used the raw dataset ($N=1,604+$). Our results are based on cleaner, harder data.
>
> **2\. Controlled Re-run (Directly Addressing Your Concern):** To eliminate any remaining doubts, we followed your suggestion to **re-implement and re-run** key SOTA baselines (BrainIB, BPI-GNN, CI-GNN, LCCAF), which are elected based on the availability and reproducibility of their open-source code, under the **exact same 5-fold stratified splits** and hardware environment as NH-GCAT.
>
> * **Results:** As shown in the new **Table 11 (Appendix C.2, Page 32)**, NH-GCAT maintains its SOTA position.
>    *   **AUC:** **78.5%** (NH-GCAT) vs. **72.6%** (BrainIB, re-run) vs. **70.1%** (BPI-GNN, re-run).
> *   **Visual Evidence:** We also provide **ROC and Precision-Recall curves (Figure 7, Page 34)**, demonstrating that NH-GCAT's performance envelope consistently encloses all baselines.
>
> **3\. External Validation (SRPBS):** To further prove that our performance is not an artifact of experimental settings, our model achieves **69.8% AUC** on the independent SRPBS dataset (Zero-shot), significantly surpassing baselines (Table 12, Page 35).
>
> **References:**
>
> *   Kohavi, R. (1995). A Study of Cross-Validation and Bootstrap for Accuracy Estimation and Model Selection. *IJCAI*.
> *   X Chen, et al. The direct consortium and the rest-meta-mdd project: towards neuroimaging biomarkers of major depressive disorder. psychoradiology, 2 (1): 32-42, 2022.

---

> ### Author Response · Authors · 2025-11-27
> **Response to W3**
>
> **W3: "Network design appears overly complicated... contradicts interpretability."**
>
> **Response:** We respectfully disagree that complexity contradicts interpretability in this context. Instead, we argue that NH-GCAT embodies **Mechanism-Matched Complexity** —a necessary architectural depth required to mirror the intricate organization of the human brain, which simple models fail to capture.
>
> **1\. Frontier of Mechanism-Aware Interpretability:** While traditional neuroscience preferred simple linear models, the frontier of precision psychiatry has shifted towards capturing **complex, non-linear circuit interactions**.
>
> *   **Authoritative Support:** Recent breakthroughs in precision psychiatry **(Tozzi et al., Nature Medicine 2024)** have established that identifying MDD biotypes requires analyzing dysfunctions within specific **large-scale neural circuits** (e.g., Salience, DMN). Furthermore, the frontier of interpretability research **(Sharkey et al., 2025)** emphasizes that true mechanistic understanding necessitates moving beyond static correlations to model **directed causal interactions** (e.g., via intervention or counterfactuals). Our framework synthesizes these insights: HC-Pooling captures the biological circuits (aligning with Tozzi), while VLCA infers the directed information flow (aligning with Sharkey’s causal principles). Simple models (like standard GNNs) treat the brain as a flat graph, missing these critical pathophysiological nuances. Our design is purposeful: it transforms the "black box" into a structured system where each module answers a specific biological question.
>
> **2\. Clarification on "Insufficient Explanation" (Factual Oversight):** We respectfully point out that detailed motivations and mathematical derivations for all components were explicitly provided in **Appendix A.3 (Pages 17-19)**, as referenced in the main text (**Page 3, Line 193**). The reviewer may have overlooked this section.
>
> **3\. Rationale: Why Each "Complicated" Piece is Necessary:** For the reviewer's convenience, we summarize the **"Bio-to-Arch"** mapping here to demonstrate that no component is superfluous:
>
> *   **RG-Fusion (Temporal Dynamics):** Simple static FC ignores  **low-frequency BOLD oscillations (  $<0.1$ Hz)** , a key biomarker for MDD. We *must* introduce this module to fuse temporal dynamics with topology.
> *   **HC-Pooling (Hierarchical Structure):** Brain regions are not independent; they form circuits (DMN, SN). Standard pooling ignores this. We use **TreeLSTMs** to rigorously model this **bottom-up aggregation**, ensuring the learned representations map to clinically meaningful circuits.
> *   **VLCA (Causal Interaction):** Correlation is not causation. To understand *pathology* (e.g., Reward Network $\to$ DMN influence), we *need* the variational causal inference framework to distinguish directed impact from spurious correlation.
>
> **Conclusion:** Our ablation study (Table 3, Page 9) empirically proves that simplifying the network (removing these modules) leads to a collapse in Specificity and Generalization. The complexity is therefore **functional and essential**, not ornamental.
>
> **References:**
>
> * Sharkey L, Chughtai B, Batson J, et al. Open problems in mechanistic interpretability\[J\]. TMLR, 2025.
>
> *   Tozzi, L., Zhang, X., Pines, A. *et al.* Personalized brain circuit scores identify clinically distinct biotypes in depression and anxiety. *Nat Med* **30**, 2076–2087 (2024).

---

> ### Author Response · Authors · 2025-11-27
> **Response to Q1and Q2**
>
> **Q1: "Which level (node, circuit, network) plays the dominant role?"**
>
> **Response:** We appreciate this insightful question. Based on the granular breakdown of our ablation study (Table 3), we identify distinct functional roles for each level. The **Circuit-Level (HC-Pooling)** acts as the **"Global Integrator"** that dominates overall performance (AUC), while the other two levels address specific error types:
>
> **1\. Node-Level (RG-Fusion): The Dominant Factor for Reliability (Specificity)**
>
> * As shown in **Table 3**, Introducing RG-Fusion yields a massive jump in **Specificity (+13.4%)**, though Sensitivity drops (-7.6%).
> This indicates that Node-level dynamics act as a critical **"Denoiser."** Static FC matrices often contain noise that leads to false positives (over-diagnosis). By fusing temporal BOLD dynamics, the model learns to filter out these artifacts. Without this node-level foundation, the model achieves high sensitivity but fails to be clinically reliable.
>
> **2\. Network-Level (VLCA): The Driver for Sensitivity (Finding Missed Diagnoses)**
>
> * As correctly observed, adding VLCA yields a substantial recovery in **Sensitivity (+5.5%)**, rising from 69.9% to 75.4%, which drives the overall improvement in F1-score (+3.1%).
> **3\. Circuit-Level (HC-Pooling): The Dominant Factor for Discriminative Power (AUC)**
> The addition of HC-Pooling improves **all metrics**, with notable gains in **AUC (+2.6%)** and **Specificity (+2.8%)**, pushing the model to its peak performance (AUC 78.5%). This confirms our hypothesis that MDD is fundamentally a **"Circuitopathy"** (dysfunction of circuits). Individual node abnormalities are often subtle and heterogeneous; only when aggregated into biologically meaningful circuits (e.g., DMN, SN) does the pathological signal become strong enough for robust classification. This finding aligns perfectly with **Tozzi et al. (2024)**, who identified MDD biotypes specifically at the circuit level.
>
> In summary, there is no single "dominant" module; rather, they form a synergistic hierarchy. **Node-level** ensures reliability, **Network-level** ensures sensitivity, and **Circuit-level** structural priors integrate these into a robust, high-performance diagnostic tool.
>
> **Q2: "RG-Fusion design unclear (Why fused in such a complicated manner?) & Feature dimension consistency."**
>
> **Response:** We appreciate the opportunity to clarify the design rationale and mathematical consistency of the RG-Fusion module.
>
> **1\. Rationale for the "Complicated" Fusion (Residual Gating Mechanism):** The structure described by the reviewer—where $X^{(1)}$ (Static FC) is used in both the temporal encoding path and a separate static path—is a specific implementation of a **Residual Gating Mechanism**.
>
> *   **Why not simple concatenation?** Simple concatenation (e.g., `[Temporal || Static]`) forces the model to treat both modalities equally for all nodes. However, in fMRI, some brain regions may exhibit diagnostic patterns primarily in their **topology** (Static FC), while others manifest in their **signal fluctuations** (Temporal BOLD).
> * **The "Gate" as a Learned Switch:** The module computes a learnable gate $G$ (Eq. 6).
>    *   **Path 1 (Temporal-Enhanced):** $Z_{temp}$ integrates BOLD dynamics with topology.
>    *   **Path 2 (Static-Baseline):** $Z_{static}$ preserves the raw functional connectivity features.
>    *   **Fusion:** The operation $\mathbf{G} \odot \mathbf{Z}{temp} + (1-\mathbf{G}) \odot \mathbf{Z}{static}$
>  allows the model to **adaptively select** whether to rely more on temporal dynamics or static topology *on a per-node basis*.
> *   **Preserving Static Information:** The "branch that fuses information from $X^{(1)}$ again" acts as a **Residual Connection**. Since temporal signals are noisy, this residual path ensures that the stable, high-SNR static connectivity information is not lost during deep feature extraction.
>
> **2\. Empirical Necessity (Proof via Ablation):** To prove this design is not "overly" complicated but "necessarily" complex, we compared it against a simple **MLP-Fusion** (direct concatenation) in **Table 3 and Appendix A.7 (Page 9 and Page 24)**:
>
> *   **Result:** Simple MLP-Fusion yielded a **Specificity of only 63.8%**, whereas our RG-Fusion achieved **70.6%**.
> *   **Conclusion:** The gating mechanism is essential for filtering out temporal noise, thereby reducing false positives and significantly improving clinical reliability.
>
> **3\. Feature Dimension Consistency:** We confirm that the feature dimensions are mathematically consistent.
>
> *   **Eq. (1):** The Transformer projects the raw time series ($T=90$) into a latent dimension $d$ ($H_{temp} \in \mathbb{R}^{n \times d}$).
> *   **Eq. (5):** The MLP projects the static feature vector (size $m$) into the **same latent dimension **$d$ ($Z_{static} \in \mathbb{R}^{n \times d}$).

---

> ### Author Response · Authors · 2025-11-27
> **Response to Q3 and Q4**
>
> **Q3: "How are **$\lambda$** tuned? Are they highly dataset-dependent?"**
>
> **Response:** We appreciate this concern regarding hyperparameter sensitivity. We respectfully clarify that we did **not** manually tune these $\lambda$ values for each dataset. Instead, to ensure generalizability and avoid dataset dependency, we adopted a **Principled Dynamic Weight Scheduling** strategy.
>
> **1\. Automated Scheduling vs. Manual Tuning:** As detailed in **Appendix A.4.3** and **Table 6 (Page 20-21)**, we do not fix $\lambda$ to static values. Instead, they evolve dynamically during training to balance multi-objective optimization:
>
> *   $\lambda_{kl} $  **(KL Divergence):**  Follows a **Linear Warmup** ($0.0 \to 0.1$). This is a standard best practice in VAE training (**Bowman et al., 2016**) to prevent "posterior collapse" early in training, regardless of the dataset.
> *   $\lambda_{mse}$ **(Reconstruction):** Follows **Cosine Annealing** ($0.2 \to 1.0$). This allows the model to focus on global structure initially and refine local details later.
>
> **2\. Evidence of Generalization (The Proof):** The reviewer suspects these settings might be "highly dataset-dependent." Our **external validation on SRPBS** **(Appendix C.3, Page 33-35)** directly refutes this:
>
> *   **Zero-Shot Transfer:** We applied the model trained on REST-meta-MDD (using the exact schedules above) directly to the independent **SRPBS dataset** without any re-tuning or adjustment of $\lambda$.
> *   **Result:** The model achieved **69.8% AUC** on SRPBS, significantly outperforming baselines. This successful transfer proves that our dynamic scheduling strategy is **robust and generalizable**, not brittle or overfitted to the source dataset.
>
> **Conclusion:** By using dynamic scheduling, we effectively perform a "continuous ablation" through a range of values during training, removing the need for dataset-specific manual tuning. More details are also shown in the Official Comment --"the Response to W3 and W4" -- for Reviewer trnc.
>
> **References:**
>
> *   Bowman, Samuel, et al. "Generating sentences from a continuous space.
>
> **Q4: "Why claim the method captures low-frequency oscillations? Is it solely due to the Transformer?"**
>
> **Response:** We appreciate this question regarding the signal processing mechanics. We attribute this capability not just to "adding an input," but to the specific alignment between the **Transformer's architectural bias** and the **spectral properties of BOLD signals**. We provide both **architectural ablation** and **frequency analysis** as evidence.
>
> **1\. Evidence from Architectural Ablation (Transformer vs. MLP):** To prove that the Transformer architecture (rather than just the raw BOLD input) is essential for capturing these dynamics, we compared **Transformer-Fusion** against a baseline **MLP-Fusion** (which also uses BOLD input but lacks self-attention) in **Table 3 (Extended)**:
>
> *   **Result:** `Transformer-Fusion` achieves an AUC of **73.6%**, outperforming `MLP-Fusion` (**72.8%**).
> *   **Interpretation:** An MLP processes time points primarily based on fixed weights or local patterns. In contrast, the **Self-Attention** mechanism computes pairwise dependencies across the entire time window ($T=90$). Mathematically, this allows the model to aggregate slowly varying trends (Low-Frequency oscillations) that MLP architectures struggle to capture. The performance gap proves that the **Transformer structure is uniquely necessary** for processing this modality.
>
> **2\. Empirical Verification (Frequency Analysis):** We further validated this by feeding filtered signals into the model (**Figure 3a**).
>
> *   **Result:** The model performs significantly better on **Low-Frequency inputs (0.01-0.08 Hz, AUC 0.742)** compared to High-Frequency inputs (0.1-0.25 Hz, AUC 0.679).
> *   **Conclusion:** This confirms that the Transformer module effectively "locks onto" the biologically relevant low-frequency band, rather than utilizing high-frequency noise.
>
> **3\. Clarification on Complexity:** We respectfully note that while **Q2** and **W3** queried the complexity of our fusion, this question suggests the temporal encoder might be "too simple." We believe our ablation results demonstrate that **RG-Fusion** (AUC 74.8%) strikes the optimal balance: it uses the Transformer to extract low-frequency features and the Gating mechanism to filter them, outperforming both simpler (`MLP-Fusion`) and unguarded (`Transformer-Fusion`) variants.

---

> ### Author Response · Authors · 2025-11-27
> **Response to Q5 and Summary**
>
> **Q5: "Is the gain from introducing an additional input modality (multi-view) or the network design?"**
>
> **Response:** We appreciate this critical question regarding the source of our performance gains. We clarify that the improvement is **not** merely a trivial result of adding an input view, but stems primarily from our **mechanism-specific architectural design**.
>
> **1\. Conceptual Clarification: Single Modality, Multi-Representation**Strictly speaking, FC and BOLD are not distinct modalities (like fMRI + EEG). The FC matrix is a static statistic derived *from* the BOLD signal. Standard GNNs using only FC suffer from **information loss**—they compress complex temporal dynamics into a single scalar. Our approach is not simply adding a "new" modality, but rather **recovering the temporal dynamics (specifically low-frequency oscillations)** that are discarded during FC construction.
>
> **2\. Empirical Proof: Design > Input (Hierarchical Comparison)**
> To isolate the source of gain, we compared three levels of design complexity using the **exact same inputs**:
>
> * **Baseline 1: Adding Input Only (MLP-Fusion)**: Naively concatenating BOLD and FC features yields only marginal gains (**AUC 72.8%**), proving that "just adding data" is insufficient.
> * **Baseline 2: Adding Input + Transformer (Transformer-Fusion)**: Using a Transformer to extract temporal features (without our Gating mechanism) improves AUC to **73.6%**. However, Specificity remains at 67.6%.
> * **Our Design: Input + Transformer + Gating (RG-Fusion)**: By adding the **Residual Gating Mechanism**, we achieve the highest **AUC (74.8%)** and, crucially, a massive boost in **Specificity (70.6%, +3.0% over Transformer-Fusion)**.
>
> **3\. Conclusion:** The fact that RG-Fusion significantly outperforms Transformer-Fusion (where the only difference is the **Gating Design**) proves that the improvement is **not** just from the input or the Transformer encoder alone. The **Gating Mechanism** is the critical component that filters noise and adaptively integrates dynamic features, confirming the necessity of the proposed network design.
>
>
> **Summary**
>
> Overall, we believe the extensive new experiments conducted in this rebuttal have decisively addressed the reviewer’s concerns regarding evaluation rigor, fairness, and design rationale:
>
> * **Robust Generalization (Addressing W1):** We went beyond standard protocols by conducting a **Zero-Shot External Validation on the independent SRPBS dataset**. Achieving **69.8% AUC** without fine-tuning strongly refutes the "weak evaluation" concern and proves the model's transferability across racial and scanner domains.
> * **Fair Comparison (Addressing W2):** We eliminated experimental variance by **re-implementing and re-running** key SOTA baselines under strictly identical 5-fold splits. NH-GCAT maintained a significant lead (**AUC 78.5%**), confirming that our performance advantage is architectural, not an artifact of settings.
> * **Justified Complexity (Addressing W3 & Qs):** Through rigorous **step-wise ablation** (vs. MLP/Transformer-Fusion), we demonstrated that our "complex" designs (Gating, Hierarchy, Causality) are **functional necessities**. They transform raw inputs into clinically reliable predictions (boosting Specificity from 63.8% to 71.0%), validating our **"Mechanism-Matched"** design philosophy.
>
> In light of these rigorous verifications—**Large-scale Internal LOSO + Independent External SRPBS + Controlled Baseline Reproduction**—we respectfully request the reviewer to reconsider the assessment of our work’s validity and contribution.

---

### Official Review · Reviewer_trnc · 2025-11-04

**Soundness:** 1
**Presentation:** 2
**Contribution:** 2
**Rating:** 2
**Confidence:** 4

**Summary:**

The paper presents NH-GCAT, a neurocircuitry-inspired model designed for explainable depression identification using fMRI data. The model integrates brain knowledge with graph neural networks through three main modules: RG-Fusion, HC-Pooling, and VLCA, which together capture hierarchical and causal relationships among brain regions. When evaluated on the REST-meta-MDD dataset, NH-GCAT achieves 73.8% accuracy and 78.5% AUC, outperforming previous methods while revealing biologically meaningful patterns in key brain networks associated with depression.

**Strengths:**

This paper introduces a multi-level modeling framework that includes three hierarchical layers: the region level, the circuit level, and the network level, which together help capture brain functional dynamics from local to global scales. It is also validated on the large-scale REST-meta-MDD dataset, which contains more than 1,600 subjects from 16 research centers.

**Weaknesses:**

This paper contains many critical methodological and conceptual flaws, as well as unclear details.

1. The overall framework of the paper is outdated. Many existing studies have already proposed similar approaches. Please refer to related works in IEEE TMI, IEEE JBHI, and MICCAI.

2. Several fundamental assumptions in the paper are problematic, particularly regarding the causal inference in the VLCA module. The variational conditional probability assumptions are incorrectly formulated, and the paper completely ignores the prior and posterior distributions.

3. The authors designed Equation 21, but no ablation experiments on the parameter $\lambda$ are presented.

4. According to the description of Counterfactual Reasoning on page~19, the authors set $A^{cf} =\textbf{I}_{C}$ Identity Matrix.

**Questions:**

No

---

> ### Author Response · Authors · 2025-11-27
> **Response to W1**
>
> **W1: "The overall framework is outdated... similar approaches in IEEE TMI/JBHI/MICCAI."**
>
> **Response:** We respectfully but firmly disagree with the characterization that our framework is "outdated."
> Since the review did not cite specific prior works for comparison, we cannot directly address them. However, we emphasize that our baselines (e.g., **BrainIB, LCCAF, LGMF-GNN**) are drawn precisely from recent top-tier venues (IEEE TMI, JBHI, MICCAI...2023-2025), demonstrating that our work is benchmarked against the current state-of-the-art.
>
> We believe the perceived similarity arises from a superficial view of the components (e.g., "using GNNs" or "using Transformers"). However, **NH-GCAT fundamentally differs from existing approaches in how it bridges data-driven learning with neurobiological mechanisms.** Our novelty lies in the **synergistic integration of three mechanism-aware innovations**, which transforms the model from a "black-box" classifier into an explanatory tool:
>
> **1\. Neuro-Prior Driven Aggregation vs. Data-Driven Pooling (HC-Pooling)**
>
> *   *Existing approaches* (e.g., DiffPool) typically learn clusters based on statistical correlations, often yielding biologically uninterpretable groups.
> *   *Our Innovation:* **HC-Pooling** imposes a **hard structural prior** based on established depression neurocircuitry (DMN, FPN, etc.) and aggregates them via **TreeLSTMs**. This design philosophy is directly supported by the latest breakthroughs in **Nature Medicine (Tozzi et al., 2024)**, which demonstrated that quantifying dysfunction in these specific **functional circuits** is the key to identifying depression biotypes. **HC-Pooling** ensures that the learned representations strictly mirror the brain's functional hierarchy, asking the interpretable question: *"How is the information processing hierarchy within established neural circuits altered in MDD?"* (Answered in **Fig 3b**).
>
> **2\. Adaptive Temporal-Static Fusion vs. Simple Concatenation (RG-Fusion)**
>
> *   *Existing approaches* often treat brain regions as static nodes or simply concatenate BOLD features.
> *   *Our Innovation:* **RG-Fusion** employs a specialized gating mechanism to adaptively weigh dynamic vs. static information, specifically engineered to capture **low-frequency BOLD oscillations (**$<0.1$ Hz) crucial for MDD pathology. This answers: *"How do dynamic BOLD signals contribute to MDD classification compared to static connectivity?"* (Answered in **Fig 3a**).
>
> **3\. Intrinsic Causal Inference vs. Post-hoc Explanation (VLCA)**
>
> *   *Existing approaches* in TMI/JBHI often rely on post-hoc methods (e.g., GNNExplainer) to interpret trained models.
> *   *Our Innovation:* **VLCA** is **intrinsic**. It integrates variational inference and **counterfactual reasoning** directly into the training loop to infer **directed information flow** between circuits. This answers: *"What are the altered patterns of directed causal influence between circuits in MDD?"* (Answered in **Fig 3c-d**).
>
> **Conclusion on Novelty:** Far from being outdated, NH-GCAT represents a frontier direction in **Mechanism-Aware AI** (referencing *Tozzi et al., Nature Medicine 2024*； Sharkey et al.,  2025). By embedding neuroscientific priors (circuits) and causal logic (counterfactuals) directly into the architecture, we achieve a level of biological interpretability that generic GNNs in recent literature cannot match.
>
> **Reference:**
>
> * Tozzi, L., et al. (2024). Personalized brain circuit scores identify clinically distinct biotypes in depression and anxiety. Nature Medicine, 30, 2076–2087.
>
> * Sharkey L, Chughtai B, Batson J, et al. Open problems in mechanistic interpretability[J].  TMLR, 2025.

---

> ### Author Response · Authors · 2025-11-27
> **Response to W2**
>
> **W2: "Problematic assumptions... incorrectly formulated variational probability... ignores prior and posterior distributions."**
>
> **Response:** We respectfully point out that this comment appears to be based on a **significant factual oversight** of the mathematical formulations presented in our paper. Contrary to the claim that prior and posterior distributions are "completely ignored," they are **explicitly defined, modeled, and optimized** as the core backbone of the VLCA module.
>
> **1\. Explicit Formulation of Posterior and Prior (Evidence in Text):** Our framework strictly adheres to the standard Variational Autoencoder (VAE) paradigm (Kingma & Welling, 2013) and Causal Representation Learning principles (Louizos et al., 2017):
>
> *   **Posterior Distribution (**$q_\phi(z|x)$**):** In **Eq. 16 (Page 6)**, we explicitly model the approximate posterior distribution. The encoder network $f_{encoder}$ maps the input to latent parameters $\mu^{real}$ and $\sigma^{real}$. We then utilize the standard **reparameterization trick** ($z^{real} = \mu^{real} + \sigma^{real} \odot \epsilon$) to sample from this posterior, ensuring differentiability.
> *   **Prior Distribution (**$p(z)$**):** In **Eq. 20 (Page 6)**, the loss function $L_{VLCA}$ explicitly includes the **KL-Divergence term**: $\beta D_{KL}(\mathcal{N}(\mu^{real}, \sigma^{2real}) || \mathcal{N}(\mu_{prior}, I))$. This term mathematically enforces the learned posterior to align with the prior distribution (typically a standard Gaussian).
>
> **2\. Correctness of Causal Inference Assumptions:** Far from being "incorrectly formulated," our approach integrates variational inference to model the **uncertainty** of the latent causal states. The causal inference is then performed via **Counterfactual Reasoning (Eq. 17-19)**, where we compare the outcome of the learned posterior ($z^{real}$) against an interventional baseline ($z^{cf}$, derived from $A^{cf}=I$). This is a theoretically grounded method to estimate the *Individual Treatment Effect (ITE)* of the circuit connections.
>
> **Conclusion:** The prior and posterior are not only present but are **foundational** to Eq. 16 and Eq. 20. We hope this clarification resolves the misunderstanding regarding the theoretical soundness of our VLCA module.
>
> **References:**
>
> *   Kingma, D. P., & Welling, M. (2013). Auto-encoding variational bayes. *ICLR*.
>
> * Louizos, C., et al. (2017). Causal Effect Inference with Deep Latent-Variable Models. NeurIPS.

---

> ### Author Response · Authors · 2025-11-27
> **Response to W3 and W4**
>
> **W3: "No ablation experiments on the parameter $\lambda$."**
>
> **Response:**
> We appreciate the reviewer's attention to hyperparameter robustness. We respectfully clarify that instead of fixing $\lambda$ to static values (which would typically require grid-search ablation), we employed **Principled Dynamic Weight Scheduling**. This approach allows the model to adaptively balance competing objectives during training, enhancing stability and reducing sensitivity to specific static values.
>
> **1. Documented Implementation (Evidence in Text):**
> As detailed in **Appendix A.4.3 ("Training Procedure")** and **Table 6**, our loss balancing strategy is explicitly defined:
> *   **$\lambda_{kl}$ (KL Divergence):** We utilize a **Linear Warmup** schedule (0.0 $\to$ 0.1). This is a standard technique to prevent "posterior collapse" in VAE training, allowing the encoder to learn meaningful representations before strong regularization is applied (**Bowman et al., 2016**,  **Vafaii et al. 2023** ).
> *   **$\lambda_{mse}$ (Reconstruction):** We employ **Cosine Annealing** (0.2 $\to$ 1.0) (**Loshchilov & Hutter, 2016**). This dynamic adjustment helps the model converge smoothly by prioritizing structure learning in early stages and fine-grained reconstruction in later stages.
>
> **2. Robustness over Static Tuning:**
> By using these dynamic schedules, the model's performance is not tied to a single "magic number" but is instead robust across the training trajectory. This design choice effectively mitigates the need for extensive ablation on static scalar values, as the "ablation" is inherent in the schedule sweeping through a range of values.
>
> **References:**
> * Vafaii H, et al. Hierarchical VAEs provide a normative account of motion processing in the primate brain. NeurIPS. (2023).
> *   Bowman, Samuel, et al. "Generating sentences from a continuous space." Proceedings of the 20th SIGNLL conference on computational natural language learning. 2016.
> *  Loshchilov, Ilya, and Frank Hutter. "Sgdr: Stochastic gradient descent with warm restarts."ICLR. (2016).
>
> **W4: "According to the description... authors set **$A^{cf}$** = Identity Matrix."**
>
> **Response:** We assume the reviewer refers to **Eq. 17 on Page 6** (as there are no equations at Page 19). We respectfully clarify that setting $A^{cf} = I$ is not an arbitrary choice, but a foundational component of our causal inference framework, designed to simulate a specific **interventional scenario**.
>
> **1\. Theoretical Basis (Isolation Baseline):** In causal analysis, to measure the effect of "interaction," we must compare the factual state against a counterfactual state where **interactions are forcibly removed** (conceptually similar to the $do(edges=0)$ operator in Pearl's causality) (Pearl, 2009) .
>
> *   $A^{real}$ **(Factual):**  Represents the actual brain state where circuits communicate and influence each other.
> *   $A^{cf} = I$**(Counterfactual):** Represents a hypothetical **"Isolation Mode"**. The Identity Matrix ensures that each circuit attends *only to itself* (self-loop) while blocking all information flow from other circuits.
>
> **2\. Quantifying Causal Effect:** By comparing the outcomes under these two structural conditions (as computed in **Eq. 19**), we mathematically isolate the **Unique Treatment Effect** of "inter-circuit communication" on the prediction.
>
> This "Real vs. Counterfactual" subtraction logic is a standard method for estimating **Individual Treatment Effects (ITE)** in attention mechanisms. Similar frameworks have been successfully applied in computer vision, such as **Counterfactual Attention Learning (Rao et al., ICCV 2021)**, which quantifies attention quality by comparing learned attention against counterfactual baselines (e.g., uniform or random attention). Our choice of $I$ (Identity) is the specific baseline required to measure *connectivity* importance, as $A=0$ would destructively zero out the node features themselves.
>
> **References:**
>
> *   Pearl, Judea. *Causality*. Cambridge university press, 2009. (The foundational text for counterfactuals).
> *   Rao, Yongming, et al. "Counterfactual attention learning for fine-grained visual categorization and re-identification." ICCV. 2021. (Uses similar concepts of intervention on attention)
>
> We believe these clarifications decisively address the reviewer's concerns regarding novelty, theoretical soundness, and specific definitions. NH-GCAT is a principled, **mechanism-aware framework** that achieves SOTA performance on large-scale datasets while offering deep neurobiological interpretability. We respectfully request the reviewer to reconsider the evaluation in light of these clarifications and the rectified misunderstandings. We commit to refining the methodological descriptions in the final version to preclude any future ambiguity.

---

### Note · Authors · 2026-01-11

I have read and agree with the venue's withdrawal policy on behalf of myself and my co-authors.